

# Does prognostic seeding along flight tracks produce the desired effects of cirrus cloud thinning?

Colin Tully , David Neubauer , Diego Villanueva , and Ulrike Lohmann

Institute for Climate and Atmospheric Science, ETH Zurich, Zurich, Switzerland

**Correspondence:** Colin Tully (colin.tully@env.ethz.ch) or Ulrike Lohmann (ulrike.lohmann@env.ethz.ch)

**Abstract.** To date the climate intervention (CI) proposal of cirrus cloud thinning (CCT) was only assessed in general circulation models (GCMs) using a globally uniform distribution of artificial ice nucleating particles (INPs). In this study, we made the first attempt using the ECHAM-HAM GCM to simulate CCT using a fully prognostic cirrus seeding aerosol species. Seeding particles were assumed to be made of bismuth-triiodide and were emitted into the atmosphere following aircraft emissions of black carbon (soot). This new approach drastically reduced the number concentration of seeding particles available as INPs in our cirrus ice nucleation sub-model compared to the globally uniform approach. As a result, we found that in order to achieve a significant signal we needed to reduce the assumed radius of emitted seeding particles by an order of magnitude to $0.01\,\mu m$ and scale the mass emissions of seeding particles by at least a factor of 100 or 1000. This latter scaling factor led to a large net TOA warming effect of $5.9\,Wm^{-2}$. This warming effect was a clear response to overseeding with a large concentration of seeding particles ($> 10^5\,L^{-1}$ in the northern hemisphere) that was most evident in the tropics. Due to this undesired effect, in a second series of simulations we avoided seeding the tropics by restricting emissions to only the northern hemisphere (NH) during winter. We also found a small and insignificant effect, or overseeding, which for the extreme case was reduced compared to the global aircraft emission scenario ($2.2\,Wm^{-2}$). Ice crystal radius anomalies were not what we expected, with the largest reduction in size found for the case with a mass scaling factor of 10 instead of the extreme, x1000, scenario. We attributed this peculiar behavior to the differences in the competition between different seeding particle concentrations and background particles. Finally, we also found that seeding with such large concentrations increased the albedo effect of mixed-phase clouds in the NH due to less efficient cloud droplet consumption, consistent with previous findings from our model. Overall, however, based on this study it is recommended to pause further modeling efforts of CCT unless more observational-based evidence of aerosol-ice-cloud interactions indicates favourable conditions for producing the desired outcome of this CI proposal.

## 1 Introduction

Cirrus cloud thinning (CCT) is a climate intervention (CI) proposal with the specific aim to enhance the outgoing flux of terrestrial, longwave (LW) radiation by counteracting the net warming effect of naturally occurring cirrus clouds (Mitchell and Finnegan, 2009; Muri et al., 2014). On average, cirrus have a weak shortwave (SW) albedo effect, allowing most incoming solar radiation to be transmitted to the lower atmosphere and the surface. They are more effective in the longwave (LW) spectrum, absorbing a significant fraction of upwelling radiation from the surface and, due to their cold temperatures, re-emitting it





with a considerably lower magnitude, thus creating a LW "trapping" effect from a top-of-atmosphere (TOA) perspective. Recent observational evidence introduces more nuance in understanding cirrus clouds' specific radiative properties, with in-situ observations showing a large local cooling potential for liquid-origin cirrus formed in "medium updraft" environments (Krämer et al., 2020). Overall, however, as cirrus consist entirely of ice, their radiative properties are dependent on their dominant ice formation mechanism.

Ice in cirrus forms either by homogeneous or heterogeneous nucleation. The former occurs as the spontaneous and rapid freezing of small aqueous solution droplets (also referred to as liquid aerosols) under the appropriate thermodynamic conditions, which are favorable at temperatures below about 235 K and at high supersaturation with respect to ice (Koop et al., 2000; Ickes et al., 2015). These conditions are closely linked to the magnitude of vertical velocity, which determines the degree of adiabatic cooling in the atmosphere. A direct relationship was found between the updraft velocity and the number of ice particles formed by homogeneous nucleation (Jensen et al., 2016b). As this process is highly dependent on the appropriate conditions, as soon as they are met rapid ice formation can occur. This results in rapid water vapor consumption, which is sparse in the upper troposphere, and as a result, limits ice crystal growth. Therefore, the ice crystal population following a spontaneous homogeneous event tends to consist of numerous small particles that produce cirrus with long cloud lifetimes (Krämer et al., 2016; Krämer et al., 2020)

Heterogeneous nucleation occurs at lower temperatures and at lower ice supersaturated conditions than homogeneous nucleation due to the energetically favorable conditions on the surface of an ice nucleating particle (INP, Kanji et al., 2017). The number of ice crystals resulting from a heterogeneous nucleation event is typically limited by the availability of INPs, which are sparsely populated at typical cirrus temperatures ($< 235$ K). However, due to this limitation and the fact that nucleation on INPs can occur at low ice supersaturation, heterogeneously-formed ice crystals can grow to larger sizes within cirrus because less of them compete for the available water vapor.

Both nucleation modes are not mutually exclusive within cirrus clouds, and understanding their complex competition for available water vapor is an area of ongoing research (Kärcher et al., 2022). However, it is understood that heterogeneous nucleation can suppress homogeneous nucleation under appropriate conditions (Lohmann and Kärcher, 2002; Kärcher and Lohmann, 2003; Lohmann et al., 2008; Mitchell and Finnegan, 2009; Kuebbeler et al., 2014; Jensen et al., 2016b, a; Kärcher et al., 2022). For example, a sufficient concentrations of INPs in an ice supersaturated environment can lead to rapid ice formation and consumption of available water vapor, which counteracts updraft-fuelled ice supersaturation increase needed for homogeneous nucleation. The effect of INPs on supersaturation was found to be less effective for larger updrafts (Kärcher and Lohmann, 2002; Kärcher et al., 2006; Jensen et al., 2016b). Overall, the ability of heterogeneous nucleation to suppress homogeneous nucleation impacts the ice crystal population, leading to fewer and larger ice crystals. This shift also impacts the radiative properties of cirrus clouds in what is known as the negative Twomey effect (Kärcher and Lohmann, 2003). With fewer and larger ice crystals present as a result of a shift from homogeneous to heterogeneous nucleation, the ice population within cirrus can more readily sediment and reduce cloud lifetimes and therefore their radiative effects (Lohmann et al., 2008). It was also found that as heterogeneous nucleation can occur at warmer temperatures, a shift towards this mode can also be realized as a cirrus cloud shift towards lower altitudes (i.e. warmer temperatures), which by itself results in weaker LW trapping




(DeMott et al., 2010). Exploiting this difference between the two ice nucleation modes is the main idea behind CCT. By using efficient artificial INPs in regions where cirrus ice formation is dominated by homogeneous nucleation, the goal is to form ice heterogeneously. The resulting ice crystals then grow rapidly to remove water vapor, a potent greenhouse gas, and sediment out of the clouds, reducing their lifetimes and subsequently their radiative effect.

Numerous modeling studies over the last decade evaluated the efficacy of CCT as a CI strategy and concluded with contrasting results. This is primarily due to the lack of a consistent approach between different modeling groups to simulate the complexities of ice formation in cirrus clouds (Gasparini et al., 2020). Early CCT studies assumed that ice in cirrus formed only by homogeneous nucleation (Storelvmo et al., 2013) and did not include pre-existing ice originating from, for example, convective detrainment (Storelvmo et al., 2013). In this case CCT produced a large cooling effect of nearly -2.0 $\mathrm{Wm^{-2}}$ by

significantly reducing homogeneous nucleation within cirrus. When heterogeneous nucleation was introduced as an ice formation source, a similar CCT efficacy was found in the CAM5 general circulation model (GCM, Storelvmo and Herger, 2014; Storelvmo et al., 2014). However, Penner et al. (2015) used the same model and included a larger concentration of background INPs, pre-existing ice crystals, and a larger spectrum of updraft velocities, and found no significant cooling in response to seeding. Similarly, Gasparini and Lohmann (2016) used the ECHAM-HAM GCM and included pre-existing ice particles in

their in-situ cirrus ice nucleation scheme. They also found no significant cooling response in their simulations that included homogeneous and heterogeneous nucleation as well as vapor deposition onto pre-existing ice crystals. Finally, Tully et al. (2022a) used a more new ice microphysics scheme that abandons heuristic ice size-class transfers and also did not find that CCT has a significant cooling potential.

Naturally, based on the findings of previous studies, CCT appears as an infeasible CI strategy on a global scale (Tully

et al., 2022a). This is in part due to the spatial availability of seeding particles. A majority of the studies described above used a globally uniform distribution of seeding particles. With highly efficient seeding particles, such an approach can lead to accumulated seeding particle impacts (i.e. "overseeding") that can produce large warming effects (Storelvmo et al., 2013; Tully et al., 2022a). Only Storelvmo and Herger (2014) and Storelvmo et al. (2014) examined non-uniform seeding particle distributions based on seasonality. They varied their seeding particle concentrations zonally, with the maximum concentrations

in the high latitudes of the winter hemisphere (north or south). In these high-latitude regions during winter, cirrus exert only a positive LW warming effect and a negligible (or zero) SW cooling effect. In addition, polar regions contain fewer background aerosols, making them more suitable for homogeneous nucleation as the dominant cirrus formation mechanism (Rogers et al., 2001; DeMott et al., 2010; Hartmann et al., 2020; Li et al., 2022). Seeding in these regions during this period may optimize CCT efficacy, which was found by both Storelvmo and Herger (2014) and Storelvmo et al. (2014) with a similar CCT cooling

effect of around -2.0 $\mathrm{Wm^{-2}}$ for their non-uniform seeding simulations as in their globally uniform cases.

The zonally-variable approach adopted by Storelvmo and Herger (2014) and Storelvmo et al. (2014) assumes a uniform distribution of seeding particle availability over specific latitude regions. Therefore, it still has the potential to overestimate the impact of seeding particles on cirrus ice nucleation competition. Mitchell and Finnegan (2009) proposed that if CCT were implemented in the real-world, a potential delivery mechanism could be to use commercial aircraft, which would have a much

less homogeneous spatial extent. To date, only Gruber et al. (2019) examined CCT based on vertical seeding particle concen-





tration profiles from aircraft emissions in a higher resolution study in a limited region over the Arctic, using the ICON-ART model (Zängl et al., 2015; Rieger et al., 2015). CCT was most effective in their simulations with no background INPs (i.e. no heterogeneous nucleation), equating to a cooling effect of nearly -7.0 $\mathrm{Wm}^{-2}$ over their region. Their CCT simulations became less effective for increasing background INP concentrations. They also note that their simulations with targeted seeding (only

seeding grid boxes if homogeneous nucleation would occur in that timestep) showed smaller ICNC reductions than seeding their entire domain. In these targeted simulations, seeding particles were effective at shutting off homogeneous nucleation in regions with suitable conditions for this ice formation process. When they seeded their entire domain, the seeding particles were injected in areas where the ice supersaturation was too low for homogeneous nucleation, thus inhibiting the development of the conditions required for this process "downstream" (Gruber et al., 2019).

Gruber et al. (2019) prescribed a homogeneous distribution of aerosols at each vertical level over their limited domain that were not removed by nucleation or sedimentation. This approach also has the potential to overestimate the impact of seeding on cirrus ice nucleation. Introducing prognostic seeding particles can address this issue as it provides the ability to trace their evolution in the atmosphere more accurately. In this study we introduce prognostic seeding particles and examine CCT efficacy based on a spatially heterogeneous distribution following aircraft emissions of black carbon (i.e. soot). We achieve this

by extending the ECHAM aerosol module, HAM, by an additional prognostic aerosol species for seeding particles as explained in Section 2. We also allow aircraft soot to act as an INP for cirrus ice nucleation by using a new soot parameterization (Section 2). Our results on the sensitivity of CCT to regional aircraft emissions and large sources of background INPs (i.e. mineral dust particles) are presented in Section 3. Finally, we conclude this study with a summary of our key findings in Section 4.

## 2 Methods

### 2.1 Model Description

We use the ECHAM6.3 atmospheric GCM (Stevens et al., 2013; Neubauer et al., 2014, 2019) coupled to the aerosol model HAM2.3 (Section 2.2, Stier et al., 2005; Zhang et al., 2012; Tegen et al., 2019). The model has a horizontal resolution of T63 (1.875 ° x 1.875 °) with 47 vertical levels (L47) up to 0.01 $\mathrm{hPa}$. The model timestep is 7.5 minutes. Monthly mean sea surface temperatures and sea ice coverage are prescribed.

Following Tully et al. (2022a), the default ECHAM two-moment ice microphysics scheme by Lohmann et al. (2007), (2M) was replaced in this study by the new Predicted Particle Properties (P3) ice microphysics scheme (Morrison and Milbrandt, 2015; Dietlicher et al., 2018, 2019). Like the 2M scheme, P3 predicts mass and number mixing ratios of various liquid and ice hydrometers, but provides an updated representation of ice microphysics by abandoning unphysical conversion rates between ice hydrometeors of different size classes (Levkov et al., 1992). Instead, ice is included under a single prognostic category that

is updated at every timestep based on mass-to-size relationships Tully et al. (2022a). This is achieved through a sub-stepping approach for prognostically solving vertical diffusion of in-cloud and precipitating hydrometeors. The P3 scheme is coupled to the new cloud fraction approach by Dietlicher et al. (2019), (D19) that allows for partial gridbox coverage of cirrus clouds above ice saturation (Dietlicher et al., 2019; Tully et al., 2022a).





Ice formation and intial growth in cirrus clouds is calculated in a separate cirrus ice nucleation competition sub-model
(Kärcher et al., 2006; Kuebbeler et al., 2014; Muench and Lohmann, 2020; Tully et al., 2022a). The scheme follows a water
vapor competition approach, whereby the available water vapor during the adiabatic ascent of a theoretical air parcel must
compete between deposition onto pre-existing ice crystals, i.e. from convective detrainment or those transported from the
mixed-phase regime, and new ice formation by heterogeneous or homogeneous nucleation. At the end of one cycle of the
cirrus sub-model, the number of new ice crystals is passed back to the cloud microphysics (P3) scheme.

Table 1 was adapted from Tully et al. (2022a) and presents a summary of the aerosol species available for heterogeneous
and homogeneous nucleation in our cirrus sub-model. Default processes (i.e. those in the base version of our model) include
deposition nucleation onto soluble (internally mixed) soot particles, deposition nucleation onto insoluble (externally mixed)
dust particles, immersion freezing of internally mixed dust particles, and homogeneous freezing of liquid sulfate particles.
Muench and Lohmann (2020) distinguish between continuous and threshold freezing processes. Continuous processes include
deposition nucleation on mineral dust particles based on the activated fraction parameterization by Möhler et al. (2006), and
on soot particles based on the "cloud-aging" parameterization by Lohmann et al. (2020). Freshly emitted externally mixed soot
particles must undergo compaction and be coated by at least a mono-layer of sulfate in order to act as INPs in our model (Mahrt
et al., 2018, 2020). For the threshold freezing processes, we assume that all particles associated with that mode nucleate ice
when the appropriate ice saturation ratio ($S_i$) conditions are met. However, as deduced from laboratory measurements, only
5 % of the available internally mixed mineral dust particles can form ice through immersion freezing (Gasparini and Lohmann,
2016).

Seeding particles made of Bismuth tri-iodide ($BiI_3$) with a density of $5778\,\mathrm{kg\,m^{-3}}$, following Mitchell and Finnegan (2009),
are included as an additional heterogeneous nucleation mode in our cirrus sub-model. The number of seeding particles available
for ice nucleation in the cirrus scheme no longer follows a globally uniform approach, and instead follows aircraft emissions
to emulate a more realistic CI scenario. As this is closely linked to their implementation as a prognostic aerosol species,
this is described in more detail in Section 2.2. However, as we include internally and externally mixed seeding particles, we
assume different ice nucleation behavior for these two particle types (Table 1). For simplicity we assume $BiI_3$ has the same
freezing properties of internally and externally mixed mineral dust particles in our model, which follow deposition nucleation
and immersion freezing processes, respectively. Following Gasparini and Lohmann (2016) and Tully et al. (2022a), the seeding
particle in our model can nucleate ice starting at a much lower critical $S_i$ threshold ($S_{i,seed}$) of 1.05 (Tab. 1). For externally mixed
seeding particles, we follow the active surface site density approach by Ullrich et al. (2017) based on Aerosol Interaction and
Dynamics in the Atmosphere (AIDA) cloud chamber experiments.

## 2.2 Prognostic Seeding Particles

The atmospheric circulation model, ECHAM, is coupled to the two-moment Hamburg Aerosol Module (HAM) for simulating
aerosol microphysics and chemistry (Stier et al., 2005; Zhang et al., 2012; Tegen et al., 2019). We use the latest version,
HAM2.3, based on Zhang et al. (2012), with updates detailed by Tegen et al. (2019). The development of HAM allows full





**Table 1.** Adapted from Tully et al. (2022a). A summary of the different aerosol species available for ice nucleation within the in-situ cirrus sub-model. We also present information on the average radius of the particles, the critical ice saturation ratio above which these particles will nucleate ice, the freezing mechanism by which nucleation will occur, and the freezing method within the context of the cirrus scheme following Muench and Lohmann (2020). "Int. mixed" stands for internally mixed (soluble) aerosol species and "Ext. mixed" stands for externally mixed (insoluble) species. Particle types (i.e. aerosol species) denoted in *italics* are included as additional processes relative to the base version of our model.

| Particle type | Mean radius ($\mu$m) | Critical $S_i$ | Freezing mechanism | Freezing method |
|---|---|---|---|---|
| Int. mixed soot | > 0.05 | Temperature-dependent, but > 1.0 | Deposition nucleation | Continuous |
| Ext. mixed dust | 0.05 to 0.5 | Temperature-dependent, but > 1.1 | Deposition nucleation | Continuous |
| | > 0.5 | Temperature-dependent, but > 1.2 | | |
| Int. mixed dust | > 0.05 | 1.3 | Immersion freezing | Threshold |
| Aqueous sulfate | All size modes: < 0.005 to > 0.5 | 1.4 | Homogeneous nucleation | Threshold |
| *Ext. Mixed seeding particles* | - | Temperature-dependent, but > 1.05 | Deposition nucleation | Continuous |
| *Int. Mixed seeding particles* | - | 1.05 | Immersion freezing | Threshold |

coupling of aerosols to the cloud microphysics scheme in order to prognostically track cloud droplet and ice crystal number concentrations (Stier et al., 2005; Lohmann et al., 2007).

In the base version of HAM, the mass mixing ratios and the number concentrations of the aerosol species, sulfate (S), black
carbon (BC), organic carbon (OC), seasalt (SS), and mineral dust (DU) are provided. Aerosol size distributions are described by the seven-mode (M7) aerosol model by Vignati et al. (2004). The scheme represents the population of aerosols following seven log-normal particle size distributions (PSDs) characterized by the number geometric mean radius ($r_g$) of the particles within the mode and their solubility. Table 2 describes the properties of the M7 modes. We extended HAM-M7 by two extra modes exclusively for internally and externally mixed seeding particles (SD) to form HAM-M9. The implementation of the
prognostic cirrus SD closely follows the methodology by Gilgen et al. (2018) for fire charcoal emissions.

Processes in HAM-M9 include aerosol emissions, aerosol microphysics, and removal processes. As these processes are described in detail by Vignati et al. (2004), Stier et al. (2005), Zhang et al. (2012), and Tegen et al. (2019), we only provide summaries in this section that are specific to our model setup and to the two extra SD modes.

Aerosol emissions are consistent with the sector-based specification described by Tegen et al. (2019) for anthropogenic
(e.g. industry, agriculture, aviation), biogenic, and fire-based sources. Anthropogenic emissions in our model use the latest Community Emissions Data System (CEDS) release for emissions for the year 2008 based on Hoesly et al. (2018) with updates provided by O'Rourke et al. (2021), namely for historic BC and OC emissions. Biogenic emissions of OC (i.e. secondary





**Table 2.** Summary of the nine log-normal modes included in HAM-M9 organized by the size class, the mode mean radius ($r_g$), and solubility (insoluble or soluble). The aerosol species are denoted as: sulfate (S), black carbon (BC), organic carbon (OC), mineral dust (DU), sea salt (SS), and seeding particles (SD). "Int. mixed" stands for internally mixed (soluble) aerosol species and "Ext. mixed" stands for externally mixed (insoluble) species. Note seeding particles mode mean radius is not defined here as we test various emissions radii in our model.

| Size class | Mode mean radius ($r_g$) | Int. mixed | Ext. mixed |
|---|---|---|---|
| Nucleation | $r_g < 0.005\,\mu m$ | S | - |
| Aitken | $0.005\,\mu m < r_g < 0.05\,\mu m$ | S, BC, OC | BC, OC |
| Accumulation | $0.05\,\mu m < r_g < 0.5\,\mu m$ | S, BC, OC, DU, SS | DU |
| Coarse | $r_g > 0.5\,\mu m$ | S, BC, OC, DU, SS | DU |
| Seed | - | SD | SD |

organic aerosols) and dimethyl sulfide (DMS) follow AeroCom-II monthly mean emissions for the year 2000 (Dentener et al., 2006). Biomass burning (fire) emissions are based on CMIP6 methodology for the year 2008 (van Marle et al., 2017). All
aerosol emissions are simulated as mass emissions in our model. Number concentrations of aerosols are obtained by following a mapping procedure that is applied to the mass emissions. The resultant number concentration is determined by the assumed radius of the emitted particle and the density of the substance, which for SD by default in our model is $0.5\,\mu m$ and $5778\,kg\,m^{-3}$ (for $BiI_3$), respectively. By changing the assumed emitted radius of a particle, one can alter the number concentration of an aerosol species in an inverse relationship as it changes how mass is distributed across the available particles (i.e. larger emitted
particles obtain lower number concentrations as individual particles can contain more mass than smaller particles). We also apply scaling factors to the mass emissions that have a direct effect on the particle number concentration.

We assume that SD originate solely from aviation sources to emulate the proposed delivery mechanism over wide areas (Mitchell and Finnegan, 2009). Their emissions follow the same spatial and temporal distribution as BC CEDS aircraft emissions. Figure 1 presents the annual vertically integrated spatial distribution of seeding particle emissions as an example. To
avoid seeding particle emissions in the mixed-phase or liquid regimes, or near the surface, we applied an online temperature filter to exclude seeding particle mass outside of the cirrus regime (T > 238 K). This approach of following flight tracks in cirrus-only conditions is a first step towards addressing the overseeding issue found by previous CCT studies that used a globally uniform seeding particle distribution (Storelvmo et al., 2013; Gasparini and Lohmann, 2016; Tully et al., 2022a). Primary emissions of SD are assigned to the externally mixed seed mode (Table 2).
SD emitted alongside aircraft exhaust will be in environments surrounded by a mixture a various aerosols, including but not limited to sulfuric acid ($H_2SO_4$) and BC (Kärcher et al., 2000; Kärcher, 2018; Durdina et al., 2019; Voigt et al., 2021). Therefore it is likely that freshly emitted SD will interact with other aerosol particles, similar to the cloud-aging of aircraft soot (i.e. BC) by sulfate found by Mahrt et al. (2018, 2020). We exclude interactions between SD and BC in our model for this study for simplicity; however, we note that this provides an opportunity for future work into the interactions between a realistic





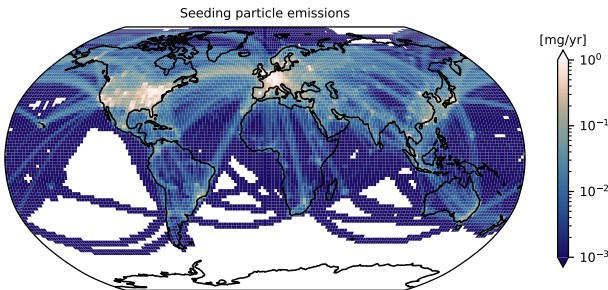

**Figure 1.** Five-year annual mean global distribution of seeding particle mass emissions (in $\mathrm{mg/yr}$) along aircraft flight routes. The mass emissions of seeding particles match those of aircraft emissions of black carbon in this case.

seeding particle material and aircraft soot, and its effect on the ice nucleation ability of both particles. We restrict interactions with SD to only sulphurous species, namely sulfate and $H_2SO_4$ gas.

Aerosol interactions are handled by the aerosol microphysics scheme, M9 based on M7 (Vignati et al., 2004; Tegen et al., 2019), and includes $H_2SO_4$- $H_2O$ droplet nucleation (Kazil et al., 2010), coagulation (Schutgens and Stier, 2014), $H_2SO_4$ condensation, and aerosol hygroscopic growth. These processes act to redistribute aerosol mass and number between the different
modes (Table 2). For SD in our model, redistribution is restricted to unidirectional transfers between only the externally and internally mixed seed mode, which can occur either by $H_2SO_4$ condensation or coagulation with nucleation mode sulfate particles (Table 2). $H_2SO_4$ condensation occurs onto all modes in M9, with accommodation coefficients of 1.0 for internally mixed modes and of 0.3 for externally mixed modes (Vignati et al., 2004; Schutgens and Stier, 2014). Coagulation can occur as an intra- or inter-modal process (Vignati et al., 2004; Schutgens and Stier, 2014). For consistency with Vignati et al. (2004) for
the externally mixed accumulation and coarse modes, we also exclude intra-modal coagulation of externally mixed SD. Instead, coagulation can occur between internally mixed nucleation mode sulfate particles and externally mixed SD. This results in a mass and number transfer to the internally mixed SD mode. Coagulation with internally mixed nucleation mode sulfate particles and internally mixed SD simply adds mass to the SD mode particles.

Removal processes include sedimentation, and dry and wet deposition. Sedimentation of aerosol particles occurs on all
model levels and is based on Stokes velocity, which accounts for particle size and density (Seinfeld and Pandis, 1998; Stier et al., 2005; Zhang et al., 2012; Tegen et al., 2019). Only particles in the larger size modes including accumulation, coarse, and seed mode can sediment in our model, consistent with HAM2.3 as presented by Tegen et al. (2019). Dry deposition is based on a deposition flux, considering the surface-type, and is calculated from the aerosol concentration, air density, and deposition velocity (Zhang et al., 2012; Tegen et al., 2019). Finally, wet deposition occurs as a scavenging process either via cloud droplet
activation or ice crystal nucleation, or by hydrometeor collection (impaction). In-cloud scavenging follows the scheme by Croft et al. (2010) that differentiates processes between convective and stratiform clouds, as well as between liquid, mixed-phase, and cirrus clouds (Tegen et al., 2019). Below-cloud scavenging is based on the size-dependent scheme by Croft et al. (2009), where aerosol particles are collected by precipitating hydrometeors (i.e. rain or snow). The scheme by Croft et al. (2010) uses a





size-dependent approach that calculates the fraction of the total ice crystal number concentration (ICNC) out of each internally
mixed size mode (excluding nucleation), assuming the largest mode (coarse in the default M7 scheme) nucleated ice crystals
first. The integration continues through subsequently smaller modes until the fraction of scavenged aerosols is less than unity.
The original version of this approach assumed ice crystals originated solely from homogeneous nucleation in cirrus. In the
cirrus regime (T < 238 K) we know this assumption no longer holds (Cziczo et al., 2013; Krämer et al., 2016; Gasparini et al.,
2018; Froyd et al., 2022). In addition, with updates made to the our cirrus sub-model by Muench and Lohmann (2020) and
Tully et al. (2022a), plus the availability of a new heterogeneous ice nucleation parameterization (e.g. aircraft soot by, Lohmann
et al., 2020), using the default nucleation scavenging scheme by Croft et al. (2010) likely under predicts the amount of aerosol
removed via wet deposition.

We updated the in-cloud nucleation scavenging scheme to account for the different sources of ice crystals in cirrus and
mixed-phase clouds, including those from homogeneous as well as heterogeneous nucleation. Our new approach for calculating
nucleation scavenging accounts for all aerosol size modes (internally and externally mixed) that are used to nucleate ice. The
fractions used to determine the number of scavenged aerosols in each mode are no longer based on the total ICNC, but rather
on the ICNC in each mode that can nucleate ice. To this end, we use the ice number tracers implemented by Dietlicher et al.
(2019) and Tully et al. (2022a). We also added two additional tracers to distinguish between ice originating from dust deposition
nucleation and dust immersion freezing in the cirrus sub-model. For simplicity in the remainder of this study, we refer to these
tracers as ice sources, and denote them as cirrus ice sources where applicable.

## 2.3 Experimental Setup

We simulated cirrus seeding using the HAM M9 model configuration as described above, and follow a similar methodology
as Tully et al. (2022a). Instead of explicitly defining seeding particle concentrations, using a globally uniform distribution both
spatially and temporally, we can alter the number concentration of seeding particles in two ways: (1) defining different sizes for
the seeding particle emissions radius and (2) scaling the mass emissions flux of seeding particles from aircraft. Both of these
methods influence the number concentration mapping procedure that is applied to the mass emissions of seeding particles in
our model (as detailed previously, Sec. 2.2). In a series of initial tests, we simulated several different seeding particle emissions
radii: 0.01, 0.02, 0.025, 0.05, 0.1, 0.25, 0.5, 1, 5, 50, and 500 μm. Compared to the globally uniform approach by Gasparini and
Lohmann (2016) and Tully et al. (2022a), using global aircraft emissions drastically reduces the number of seeding particles
available for ice nucleation in cirrus (Appendix A). As seeding particle number concentrations are indirectly related to the
emissions radius, the largest number concentration of seeding particles ($<100\,L^{-1}$), corresponded to an emissions radius of
0.01 μm. This led to a small and insignificant net TOA anomaly ($0.001 \pm 0.91\,Wm^{-2}$) as well as small ICNC anomalies
relative to the reference case with no seeding particle emissions. For all other simulations with increasing seeding particles
emissions radii, we found similarly insignificant signals.

For that reason, the results presented in this study use a combination of the two methods to alter seeding particle number
concentration. We tested three different seeding particle emissions radii (0.01, 0.1, and 1 μm), and also applied a mass emissions
scaling factor of one (i.e the mass emissions were identical to those of aircraft BC), 10, 100, and 1000 for a total of 12





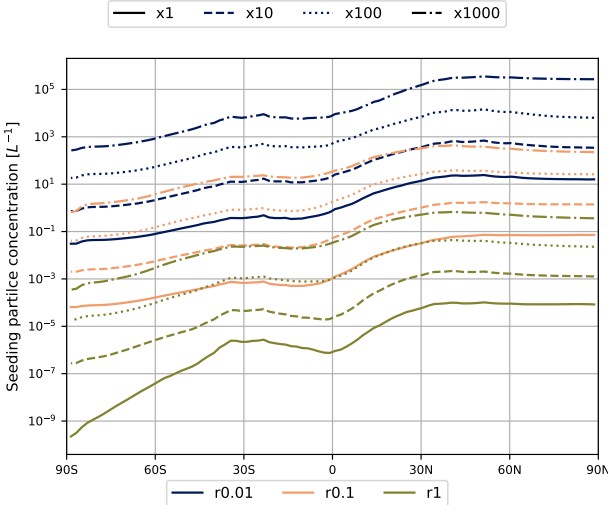

**Figure 2.** Five-year zonal mean seeding particle concentrations (in $L^{-1}$). Each color represents one of the three emissions radii: 0.01 μm (dark blue), 0.1 μm (orange), and 1 μm (olive green). Emissions mass scaling factors are represented by the line-style: x1 (solid), x10 (dashed), x100 (dotted), and x1000 (dot-dashed).

simulations. Figure 2 presents the zonal mean of the seeding particle concentrations (in $L^{-1}$) that are used as an input variable to our cirrus sub-model for the three different radii and four different mass scaling factors we tested. As noted above, there is

an inverse relationship between the size of the emitted particle and their number concentration. The concentration of seeding particles with an emissions radius of 1 μm never exceed about $1\,L^{-1}$. Meanwhile, we find the largest concentration ($> 10^5\,L^{-1}$) for the case with an emission radius of 0.01 μm and a mass scaling by a factor of 1000. Seeding particle concentrations are higher in the northern hemisphere (NH) as this coincides with the greatest aircraft emissions (i.e. the heaviest air traffic corridors).

Each simulation was conducted for five years between 2006 and 2010, including three months of spin-up from 1st October 2005. We follow Tegen et al. (2019) and run nudged simulations that relax modelled prognostic variables, surface pressure, vorticity, and divergence (Jeuken et al., 1996) toward ERA-Interim reanalysis data (Berrisford et al., 2011; Dee et al., 2011). Sea surface temperatures and sea ice coverage are based on observed monthly mean data by Atmospheric Model Intercomparison Project (AMIP) simulations (Hurrell et al., 2008). Aerosol emissions are from the year 2008, following the methodology as

described in Section 2.2.

Finally, we determine significance using the false discovery rate method by Wilks (2016) that accounts for high spatial correlation between neighboring grid points in independent t tests. Following Tully et al. (2022a) we calculate a 5 % significance.





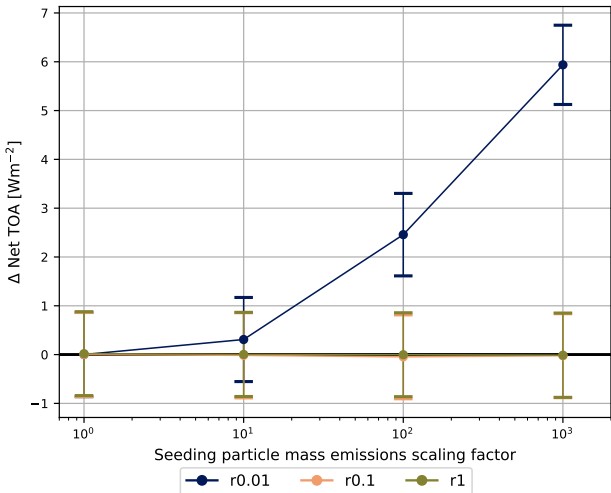

**Figure 3.** Five-year annual global mean net TOA radiative anomalies (in $\mathrm{Wm^{-2}}$) for the each seeding particle emissions mass scaling factor for the scenarios with global aircraft emissions. Each color represents one of the three emissions radii: 0.01 µm (dark blue), 0.1 µm (orange), and 1 µm (olive green). The error bars indicate the 95 % confidence interval around the five-year mean.

## 3 Results and Discussion

### 3.1 Global aircraft seeding

Global mean net top-of-atmosphere (TOA) and net cloud radiative effect (CRE) anomalies scale with the number concentration of seeding particles (Fig 3 and Tab. 3). As expected, we find the largest positive net TOA anomaly when seeding with the largest average seeding particle number concentration ($> 10^5 \, \mathrm{L^{-1}}$, Fig. 2) that is associated with the case with a mean emissions radius of 0.01 µm (r0.01) and a mass scaling factor of 1000 (high-seeding). The large TOA anomalies are driven by a large increase in the LW cloud radiative effect (CRE) by $10.1 \, \mathrm{Wm^{-2}}$ (Tab. 3), indicating a significant change in cirrus cloud properties. This

is partially offset by an increase in the magnitude of the SW CRE that slightly exceeds the total TOA SW anomaly (Tab. 3). To some extent this could be linked to optically thicker cirrus (Krämer et al., 2020), or optically thicker lower-lying mixed-phase or liquid clouds (Twomey, 1959, 1977; Albrecht, 1989). The positive TOA anomaly is reduced by over 50 % when reducing the mass scaling to 100 (r0.01 mid-seeding), but is still large and significant. It is also driven by large positive LW CRE anomalies. For all other seeding cases, the net TOA anomaly is uncertain on a 95 % confidence level. Comparing Tab. 3 to Fig. 2, it appears

that in our model in order obtain a significant radiative signal in response to seeding, the average number concentration must exceed $1000 \, \mathrm{L^{-1}}$, which occurs for only two cases: r0.01 mid and high-seeding. The response we find is only positive (i.e. warming) when applying seeding particles using global aircraft emissions.

Figure 4 presents five-year annual zonal mean profiles of cloud fraction, cirrus ice sources, and ice water content (IWC) for our case with no seeding particles (first column) and the anomaly relative to our reference case for the r0.01 high-seeding case





**Table 3.** Five-year global mean net TOA and net CRE radiative balance anomalies in $\mathrm{Wm}^{-2}$, as well as their SW and LW components for each of the seeding particle emission radii tested for the global aircraft seeding scenario. Each quantity includes the 95 % confidence interval equating to 2 standard deviations of the mean values of the 5-year data sets. Values in bold denote those that are statistically distinct from zero based on the 95 % confidence level.

| Seeding particle emission radius | net TOA | TOA SW | TOA LW | net CRE | SWCRE | LWCRE |
|---|---|---|---|---|---|---|
| μm | | | x1 | | | |
| 0.01 | 0.00 ± 0.91 | -0.04 ± 0.61 | 0.04 ± 0.34 | 0.13 ± 0.78 | 0.08 ± 0.81 | 0.05 ± 0.14 |
| 0.1 | 0.00 ± 0.91 | 0.01 ± 0.62 | -0.01 ± 0.34 | 0.00 ± 0.78 | 0.02 ± 0.81 | -0.01 ± 0.13 |
| 1 | 0.02 ± 0.91 | 0.03 ± 0.61 | -0.01 ± 0.34 | 0.02 ± 0.78 | 0.02 ± 0.80 | -0.01 ± 0.13 |
| | | | x10 | | | |
| 0.01 | 0.31 ± 0.91 | -0.37 ± 0.61 | **0.68 ± 0.34** | 0.60 ± 0.77 | -0.18 ± 0.81 | **0.78 ± 0.14** |
| 0.1 | -0.02 ± 0.92 | -0.01 ± 0.61 | -0.01 ± 0.34 | 0.00 ± 0.79 | 0.01 ± 0.81 | -0.01 ± 0.13 |
| 1 | 0.00 ± 0.91 | 0.01 ± 0.61 | 0.00 ± 0.34 | 0.00 ± 0.79 | 0.01 ± 0.81 | -0.01 ± 0.14 |
| | | | x100 | | | |
| 0.01 | **2.46 ± 0.90** | **-2.34 ± 0.58** | **4.80 ± 0.36** | **2.57 ± 0.77** | **-2.22 ± 0.76** | **4.79 ± 0.13** |
| 0.1 | -0.05 ± 0.90 | -0.04 ± 0.61 | -0.01 ± 0.34 | 0.03 ± 0.78 | 0.01 ± 0.81 | 0.02 ± 0.13 |
| 1 | 0.00 ± 0.91 | 0.01 ± 0.61 | -0.02 ± 0.34 | 0.00 ± 0.78 | 0.01 ± 0.80 | -0.01 ± 0.13 |
| | | | x1000 | | | |
| 0.01 | **5.94 ± 0.86** | **-5.05 ± 0.56** | **10.99 ± 0.36** | **5.04 ± 0.72** | **-5.06 ± 0.75** | **10.10 ± 0.17** |
| 0.1 | -0.02 ± 0.90 | -0.26 ± 0.60 | 0.24 ± 0.34 | 0.19 ± 0.78 | -0.17 ± 0.81 | 0.36 ± 0.13 |
| 1 | -0.01 ± 0.91 | 0.01 ± 0.62 | -0.03 ± 0.33 | -0.01 ± 0.78 | 0.01 ± 0.81 | -0.02 ± 0.13 |

(second column). There is a notable dipole structure in the high-level cloud fraction anomaly in response to the high concentration of seeding particles, similar to the findings by Tully et al. (2022a), with cirrus dissipation in the troposphere (between the black and blue-dashed lines in Figure 4b) and a higher cloud fraction in the lowermost stratosphere. There are also positive low-level cloud fraction anomalies in the tropics and mid-latitudes of both hemispheres (discussed below). In the troposphere, this cloud fraction behavior is due to a large shift in ice formation mechanisms in cirrus from homogeneous to heterogeneous

nucleation (Figure 4d and f). In this case the cirrus ice source from heterogeneous nucleation (ICNC HET, Fig., 4e) refers to the sum of all background processes, including deposition nucleation onto externally mixed mineral dust, immersion freezing of internally mixed dust, and freezing of internally mixed soot particles. For the seeding cases, deposition nucleation and immersion freezing of externally and internally mixed seeding particles, respectively, are added to the background heterogeneous nucleation processes in Fig. 4f.

A majority of ice in cirrus in our model originates from homogeneous nucleation (ICNC HOM) in the unseeded reference case (Fig. 4c). By adding a large concentration of seeding particles, homogeneous nucleation is almost entirely shut off in most





regions and is replaced by a larger number of ice crystals that originate from heterogeneous nucleation in both the troposphere and the lower stratosphere ($> 1000\,\mathrm{L}^{-1}$ towards the NH high latitudes, Fig. 4d). Ice source anomalies for heterogeneous nucleation show that seeding particles overtake background heterogeneous nucleation processes on mineral dust and soot

particles in the troposphere (Appendix B). As noted above, this shift in nucleation mode dominance leads to the noticeable reductions in cloud fractions in the troposphere, which is an artefact of the cloud fraction scheme we use in the model (Tully et al., 2022a). The new cloud fraction scheme by Dietlicher et al. (2019) assumes that cirrus fully cover a gridbox if the grid-mean relative humidity (RH) is sufficiently high for aqueous solution droplets to nucleate ice homogeneously according to Koop et al. (2000). The large shift to heterogeneous nucleation in cirrus reduces RH values by nearly $10\,\%$ (not shown) in

the same areas where we find negative cloud fraction anomalies, thus preventing the sufficiently high RH values needed for homogeneous nucleation and for full gridbox coverage of cirrus. This response is in-line with the intention of CCT, but the positive cloud fraction anomalies in the stratosphere counteract this intended cirrus thinning (discussed below).

The large injection of seeding particles also results in an extensive positive IWC anomaly in the cirrus regime in the troposphere and in the lower stratosphere (Figure 4h). Combined with the large increase in ICNC from the shift of homogeneous to

heterogeneous nucleation, this indicates a large reduction in the size of ice crystals. Vertical mean anomaly profiles of the mean ice crystal effective radius show that this is the case. Ice crystals are reduced in size by nearly $4\,\mu\mathrm{m}$ for our r0.01 high-seeding case between 200 and $300\,\mathrm{hPa}$ in the tropics and roughly 300 and $400\,\mathrm{hPa}$ in the NH (Fig. 5a and b). While the discussion here will focus on the anomalies for the r0.01 high-seeding case, we find the most notable reduction in ice crystal size for the r0.01 mid-seeding case in the tropics ($6\,\mu\mathrm{m}$) and in the r0.01 low-seeding case (with mass emissions scaling factor of 10)

in the NH ($> 4\,\mu\mathrm{m}$), which is unexpected and will be discussed further below. We also find that the ice radius anomalies for the r0.1 high-seeding case (emissions radius of $0.1\,\mu\mathrm{m}$) are similar to those for the r0.01 low-seeding case (though slightly weaker in the tropics) as the seeding particle concentrations in these two cases are similar (Fig. 2). For the r0.01 case with no emissions scaling (factor of one), the ice crystal radius anomaly is negligible in the tropics, whereas it shows that ice crystals are reduced in size by nearly $3\,\mu\mathrm{m}$ in the NH. Seeding with an emissions radius of $0.1\,\mu\mathrm{m}$ appears to lead to noticeable changes

in ice crystal radius for only the high-seeding case in the tropics, and the high and mid-seeding cases in the NH. All other ice radius anomalies for the other tested seeding particle emissions radii and mass scaling factors are negligible. Note that southern hemisphere (SH) anomalies are not shown due to the relatively low aircraft emissions in this region (Fig. 1).

In the tropics, while we find a decrease in ice crystal size at higher levels, there is a small increase in the mean ice crystal radius by almost $1\,\mu\mathrm{m}$ at $400\,\mathrm{hPa}$ in the r0.01 high-seeding case. Here the injection of a large concentration of seeding particles

that can nucleate ice at relatively low $S_i$ (1.05) forms some ice crystals that rapidly grow and sediment. However, the number of these large ice crystals is reduced relative to the unseeded reference case as the IWC at lower levels ($p > 300\,\mathrm{hPa}$) decreases by up to $-10\,\mathrm{mg.m}^{-3}$ (Figure 4h). Nevertheless, the main effect we find in the tropics is the formation of a large number of smaller ice crystals that remain aloft. We do find a large heating in this region by more than $5\,\mathrm{K}$ (Figure 6), which is due to a large increase in LW heating (up to about $0.8\,\mathrm{K/day}$, not shown) from the fewer, but optically thicker cirrus in this region. This

results in tropospheric stabilization and a decrease in the global mean convective precipitation rate in our model by roughly $0.26\,\mathrm{mm/day}$. A warmer tropical troposphere also means that a larger amount of water vapor can be transported into the





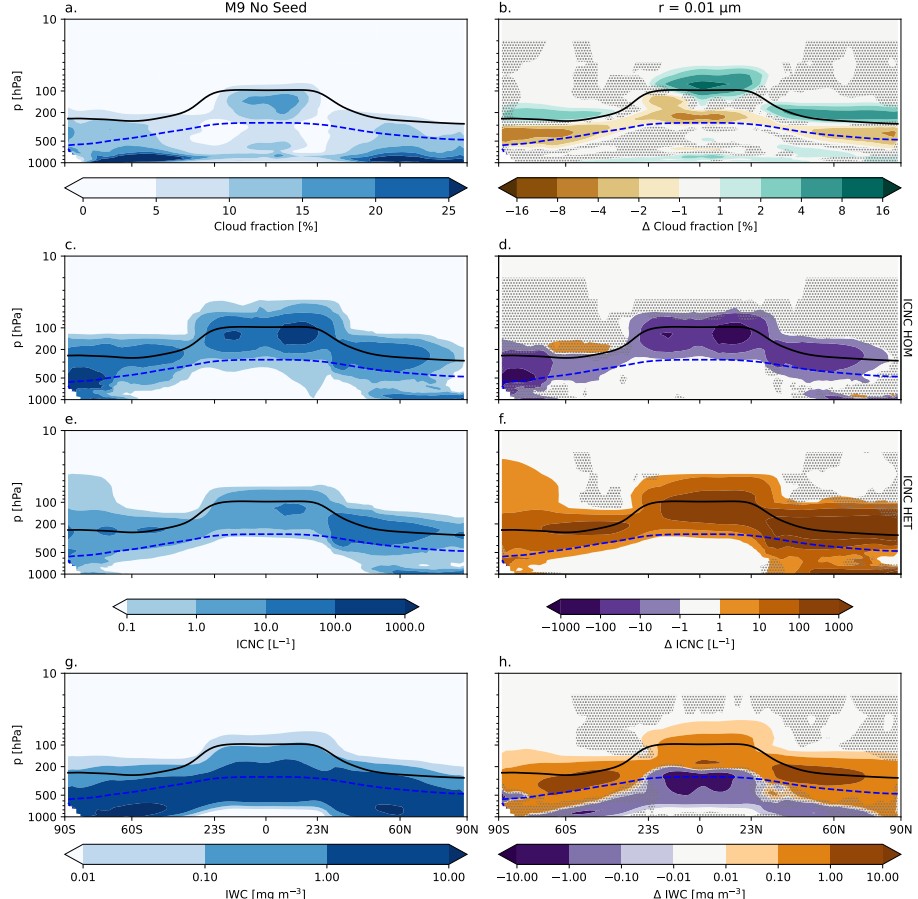

**Figure 4.** Five-year zonal mean (a) cloud fraction in %, in-situ cirrus ice sources in $L^{-1}$ for ice originating from (c) homogeneous nucleation (ICNC HOM) and (e) heterogeneous nucleation (ICNC HET), and IWC in $mg.m^{-3}$ for the unseeded reference case in the first column. The respective anomalies for the seeding case with an emissions radius of $0.01\,\mu m$ and a mass emissions scaling factor of 1000 are presented in the second column. ICNC HET refers to the sum of all the heterogeneous nucleation sources in our cirrus scheme, including deposition nucleation onto externally mixed mineral dust, immersion freezing of internally mixed mineral dust and soot particles, and in the case that seeding is active deposition nucleation and immersion freezing of externally and internally mixed seeding particles, respectively. The stippling denotes insignificant data points at the 95% confidence level according to the independent t-test controlled by the false discovery rate method.

stratosphere. In fact, for this extreme case we found that the specific humidity increases by as much as $10\,mg/kg$ in the lower stratosphere. This, combined with the availability of a larger number of seeding particles contributes to the large positive cloud fraction anomalies in the lowermost tropical stratosphere of at least $8\,\%$ in a region that was previously sparsely populated by cloud (Figure 4a). This also explains the small positive ice crystal radius anomaly at high altitudes ($p < 100\,hPa$) in the tropics (Fig. 5a).



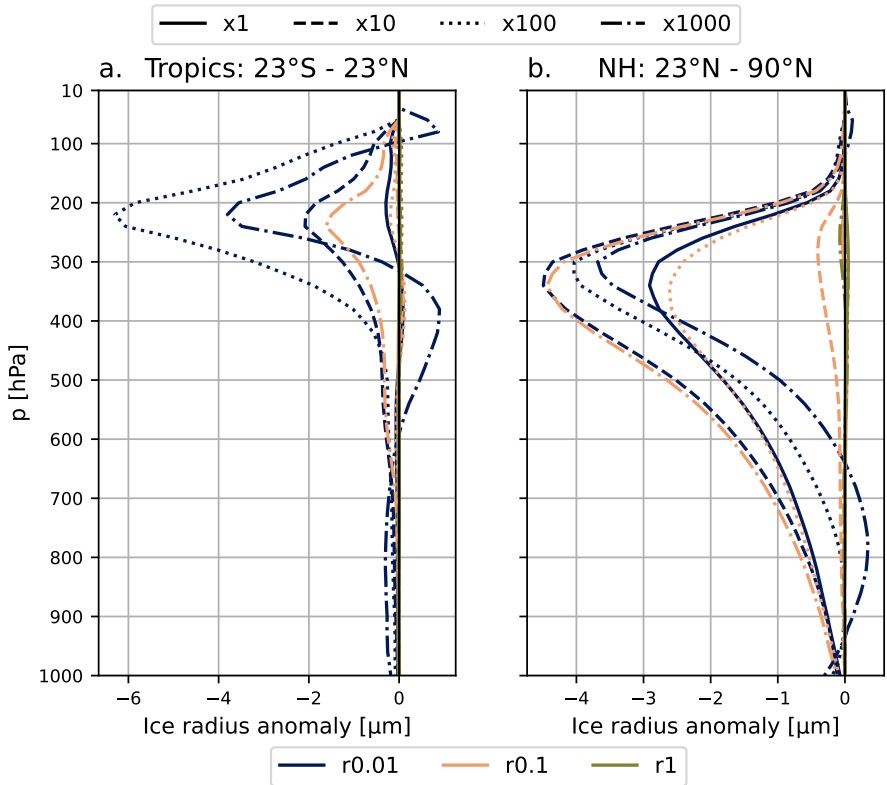

**Figure 5.** Five-year vertical mean ice crystal effective radius anomalies (in μm) for the 12 different global seeding cases tested in the model for (a) the tropics between 30 °S and 23 °N, and (b) the NH between 23 °N and 90 °N. Each color represents one of the three emissions radii: 0.01 μm (dark blue), 0.1 μm (orange), and 1 μm (olive green). Emissions mass scaling factors are represented by each line-style: x1 (solid), x10 (dashed), x100 (dotted), and x1000 (dot-dashed). Note the different scale of the x-axes.

Meanwhile, in the NH we find small positive ice crystal radius anomalies in lower levels ($p > 600\,\mathrm{hPa}$) for the r0.01 high-seeding case that is consistent with larger ice crystals that sediment more readily. This may be the case in some regions of the NH where we find positive ICNC HET anomalies up to $1000\,\mathrm{L}^{-1}$ at lower levels and a reduction of IWC up to $1.0\,\mathrm{mg.m}^{-3}$. 
In the Arctic we find the opposite case with positive IWC anomalies at low levels up to $1.0\,\mathrm{mg.m}^{-3}$. However, these low-level IWC signals are insignificant as indicated by the stippling in Fig. 4h. Seeding in this region appears to have two effects. As noted above, ice nucleation by the high concentration of seeding particles is able to overtake background heterogeneous nucleation processes on mineral dust and black carbon particles. On the one hand, this leads to rapid consumption of water vapor by some of the ice crystals that form on the seeding particles followed by rapid growth and a small enhancement of 
sedimentation. On the other hand, such rapid ice crystal growth does not leave much water vapor for the remaining ice crystals, which impedes their growth. The larger ICNC that formed by heterogeneous nucleation leads to fewer and optically thicker cirrus. This directly influences the large positive LW CRE we find for the r0.01 high-seeding case (Fig. 3 and Tab. 3).



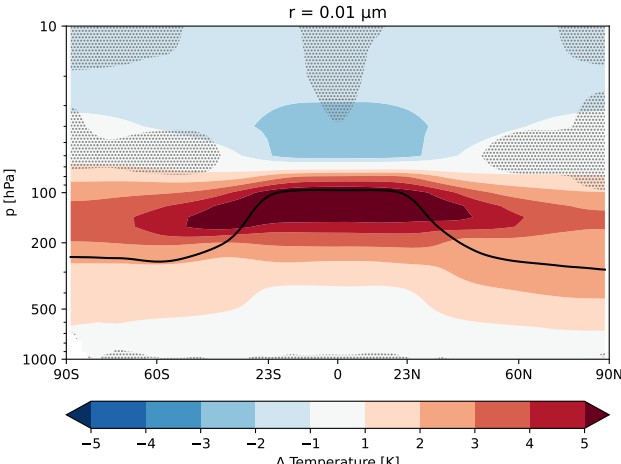

**Figure 6.** Five-year zonal mean temperature anomaly (in K) for the case with seeding particles with a mean emissions radius of 0.01 μm and a emissions mass scaling factor of 1000. The black denotes the WMO-defined five-year annual mean tropopause. The stippling denotes insignificant data points on the 95% confidence level according to the independent t-test controlled by the false discovery rate method.

A couple of factors contribute to the positive cloud fraction anomalies in the lowermost stratosphere in the NH outside of the tropics (Figure 4b). In the unseeded reference case cloud fractions in the lower stratosphere are at most 10 %. Even though there

are ice crystals present that originate from both homogeneous and heterogeneous nucleation, their concentrations do not exceed $100\,\mathrm{L^{-1}}$, except in convective outflow regions in the tropics. Like in the troposphere, the majority of ice crystals originates from homogeneous nucleation. These ice crystals also contribute very little to the overall mass of ice in the atmosphere (Figure 4g). When the large concentration of seeding particles is added in our r0.01 high-seeding simulation, we also find a shift in ice nucleation mechanism to heterogeneous nucleation by more than $1000\,\mathrm{L^{-1}}$, associated with relatively large increases in ice

mass by up to $10\,\mathrm{mg.m^{-3}}$. Such a large influx of seeding particles in previously sparsely populated regions therefore contributes to new cloud formation. Higher temperatures may prevent high ice supersaturation required for homogeneous nucleation from developing, which also contributes to a larger fraction of heterogeneous nucleation on background mineral dust particles that reaches these levels (Appendix B). This compounds the effect of the high seeding particle concentration. Therefore, the new cirrus formation in this region also contributes to the large positive LW CRE, net CRE, and net TOA anomalies we find for this

seeding scenario (Tab. 3).

The small positive cloud fraction anomalies at lower altitudes in the tropics and NH likely occur due to diminished ice sedimentation efficiency that reduces mixed-phase processes such as riming (Borys et al., 2003; Waitz et al., 2022) and the Wegener–Bergeron–Findeisen (WBF) process (Wegener, 1911; Bergeron, 1935; Findeisen et al., 2015; Storelvmo and Tan, 2015), thus fewer cloud droplets are consumed, leading to longer-lived low clouds. In fact, throughout the NH the liquid water

content (LWC) of lower lying clouds increases by at most $4\,\mathrm{mg.m^{-3}}$ (not shown). In the tropics this could also be linked to tropospheric stabilization that transports less liquid water to higher altitudes.





It is clear that injecting a large number of seeding particles leads to undesirable effects in our model, with fewer but optically thicker clouds in the troposphere and new cloud formation in the stratosphere. The former effect is an artefact of our cloud fraction parameterization, and could be addressed by using an updated method that accounts for the distinction between in-
cloud and cloud-free water vapor (Muench and Lohmann, 2020). What remains uncertain is why we find larger reductions in average ice crystal size when seeding with lower concentrations, which will be examined in more detail in the next section. The main outcome of global aircraft seeding is the large impact on the tropics. Heterogeneous nucleation onto numerous seeding particles in our r0.01 high-seeding case replaces ICNC HOM and leads to troposhereic stabilization, thus reducing convective precipitation. Overall, these effects strengthen the case that cirrus seeding efforts should not target tropical regions (Storelvmo
and Herger, 2014; Gasparini et al., 2017).

## 3.2   Northern hemisphere-only wintertime seeding

In order to avoid seeding the tropics we conducted another series of simulations with the same particle sizes and mass scaling factors described previously, but with geographically restricted emissions to only the NH between 23 °N and 90 °N. We chose 23 °N as our southernmost boundary for seeding particle emissions as this latitude corresponds to the Tropic of Cancer. How-
ever, we restricted seeding particle emissions further by only seeding during NH wintertime (November-February) as this was suggested to optimize cirrus seeding efficacy (Storelvmo and Herger, 2014; Storelvmo et al., 2014).

Tab. 4 presents the five-year annual global mean net TOA and CRE radiative anomalies as well as their SW and LW components for all 12 NH winter seeding cases. Radiative anomalies for the period between November and February are presented in Appendix C. Compared to global aircraft seeding (Tab. 3) we find a reduction in the positive TOA anomaly by roughly 63 %
when seeding only the NH during winter for the extreme case, r0.01 high-seeding, but the response remains large. As before, the TOA response is driven by cloud effects, with the net CRE accounting for roughly 99 % of the net TOA response and even exceeding the net TOA response for the other cases with r0.01 and different mass scaling factors, indicating rapid cloud adjustments for these latter cases. The positive (warming) effects are certain for the r0.01 high and mid-seeding cases with NH wintertime seeding. There is a lack of certainty in the radiative response on the 95 % level for all other cases. For example, for
the r0.01 low-seeding (scaling factor of 10) case, a negative (cooling) response to the lower seeding particle concentration is within the range of uncertainty. In the rest of this section we restrict ourselves to examining the microphysical responses to the r0.01 NH wintertime seeding cases.

Fig. 7 presents the NH wintertime zonal mean anomalies for four different parameters for the r0.01 low, mid, and high-seeding cases. Each plot includes the 95 % confidence interval around the mean anomaly for each latitude. Like the TOA
radiative anomalies in Tab. 4, the anomalies of cloud properties scale with the number concentration of seeding particles, which is highest for the x1000 case. For this extreme case, positive cloud fraction anomalies by at most 10 % contribute to the large positive TOA anomalies (Tab. 4). This results in a zonal average warming in the NH by about 1 K (not shown). Cloud fraction anomalies are smaller for the other two cases in Fig. 7a. For the low-seeding case it is uncertain whether seeding leads to higher cloud fractions and warmer temperatures, in line with the uncertain TOA radiative anomalies (Tab. 4).



**Table 4.** Five-year global mean net TOA and net CRE radiative balance anomalies in $\mathrm{Wm}^{-2}$, as well as their SW and LW components for each of the seeding particle emission radii tested for the NH wintertime seeding scenario. Each quantity includes the 95 % confidence interval equating to 2 standard deviations of the mean values of the 5-year data sets. Values in bold denote those that are statistically distinct from zero based on the 95 % confidence level.

| Seeding particle emission radius | net TOA | TOA SW | TOA LW | net CRE | SWCRE | LWCRE |
|---|---|---|---|---|---|---|
| µm | x1 | | | | | |
| 0.01 | 0.02 ± 0.90 | -0.01 ± 0.61 | 0.03 ± 0.33 | 0.07 ± 0.78 | 0.04 ± 0.81 | 0.03 ± 0.14 |
| 0.1 | 0.01 ± 0.91 | 0.01 ± 0.61 | 0.00 ± 0.34 | 0.00 ± 0.78 | 0.01 ± 0.80 | 0.00 ± 0.13 |
| 1 | 0.00 ± 0.91 | 0.00 ± 0.60 | 0.00 ± 0.35 | 0.00 ± 0.78 | 0.00 ± 0.80 | 0.00 ± 0.13 |
| | x10 | | | | | |
| 0.01 | 0.22 ± 0.90 | -0.09 ± 0.61 | 0.31 ± 0.34 | 0.33 ± 0.77 | -0.02 ± 0.80 | **0.35 ± 0.14** |
| 0.1 | 0.01 ± 0.91 | 0.02 ± 0.61 | -0.01 ± 0.34 | 0.01 ± 0.78 | 0.02 ± 0.80 | -0.01 ± 0.14 |
| 1 | 0.01 ± 0.91 | 0.01 ± 0.60 | 0.01 ± 0.34 | 0.00 ± 0.78 | 0.01 ± 0.80 | -0.01 ± 0.13 |
| | x100 | | | | | |
| 0.01 | **1.04 ± 0.89** | -0.47 ± 0.60 | **1.51 ± 0.33** | **1.19 ± 0.76** | -0.41 ± 0.80 | **1.61 ± 0.15** |
| 0.1 | 0.01 ± 0.91 | 0.01 ± 0.61 | 0.00 ± 0.34 | 0.00 ± 0.78 | 0.01 ± 0.80 | 0.00 ± 0.13 |
| 1 | 0.00 ± 0.91 | 0.00 ± 0.60 | 0.00 ± 0.35 | 0.00 ± 0.78 | 0.00 ± 0.80 | 0.00 ± 0.13 |
| | x1000 | | | | | |
| 0.01 | **2.19 ± 0.91** | **-1.37 ± 0.61** | **3.56 ± 0.35** | **2.17 ± 0.77** | **-1.36 ± 0.79** | **3.52 ± 0.13** |
| 0.1 | 0.06 ± 0.90 | -0.04 ± 0.60 | 0.10 ± 0.34 | 0.10 ± 0.77 | -0.01 ± 0.80 | 0.11 ± 0.13 |
| 1 | 0.00 ± 0.91 | 0.01 ± 0.61 | -0.01 ± 0.34 | 0.00 ± 0.78 | 0.01 ± 0.80 | -0.01 ± 0.14 |

The differences in ice property anomalies between the three cases are interesting. Similar to our global aircraft seeding scenarios, the NH wintertime zonal mean ice crystal effective radius anomalies show unexpected behavior, with the largerst reduction in zonal average ice crystal radius found for the r0.01 low-seeding case (Fig. 7b). The average ice crystal size for this case is reduced by roughly 2 µm, followed by the mid-seeding case. The ice crystal radius anomaly for the high-seeding case is much smaller, with some regions showing a slight positive signal, but it is highly uncertain. This peculiar response

is the opposite of what we would expect and may be explained by the zonal mean anomalies for the cirrus ice sources for homogeneous and heterogeneous nucleation ICNC HOM and ICNC HET (Fig. 7c and d).

We find that in the high-seeding case ICNC HOM is reduced throughout the NH (Fig. 7c), whereas seeding in the other two cases has a relatively negligible and uncertain effect. All cases show positive ICNC HET anomalies. The signal varies by at least one order of magnitude for each mass scaling factor (Fig. 7d). This is due to the large differences in concentration

of seeding particles available in the cirrus scheme between these three cases. Fig. 2 shows that the concentration of seeding particles in the NH decreases from $> 10^5 \, \mathrm{L}^{-1}$ for the high-seeding case to $< 1000 \, \mathrm{L}^{-1}$ for the low-seeding case. Note, that





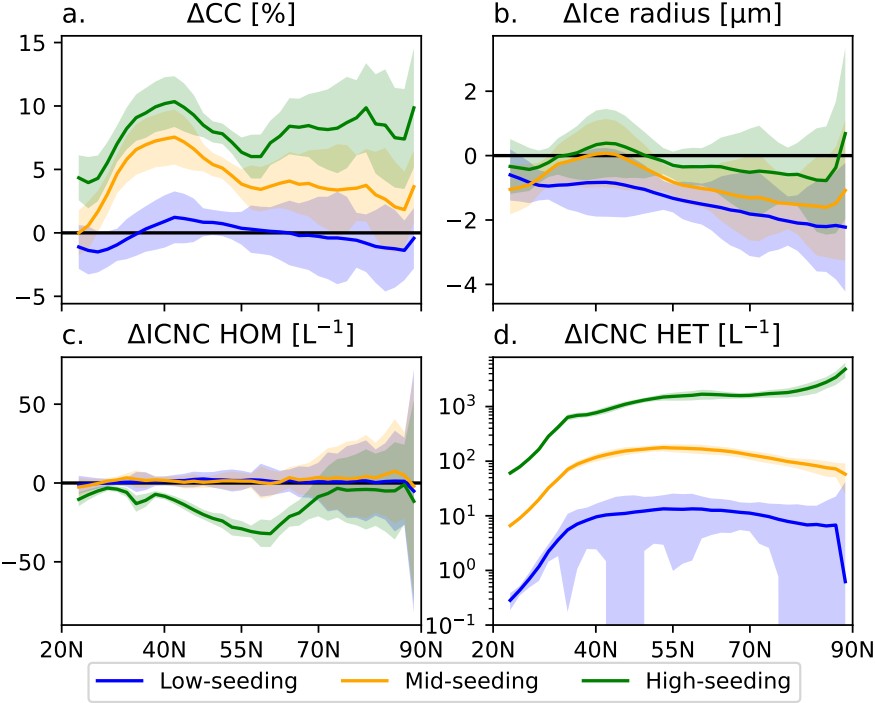

**Figure 7.** Five-year northern hemisphere winter zonal mean anomalies for (a) cloud fraction in %, (b) temperature in K, and the in-situ cirrus ice sources in $L^{-1}$ for ice originating from (c) homogeneous nucleation and (d) total heterogeneous nucleation as the sum of processes on background mineral dust, soot, and seeding particles. The anomalies are shown for the three seeding cases with particles with a mean emissions radius of 0.01 μm with a emissions mass scaling factor 10 (low-seeding, blue), 100 (mid-seeding, orange), and 1000 (high-seeding, green). The shading denotes the 95 % confidence interval around the mean value at each latitude. Note: the uncertainty for the r0.01 low-seeding ICNC HET in subplot d is large enough that is extends to values < 0, which cannot be displayed on the y-axis log scale.

while this figure shows zonal mean seeding particle concentrations for the global seeding simulations, with the NH wintertime filtering applied we find that the concentrations for all 12 combinations of seeding particle emissions radius and mass emissions scaling factor are similar in these regions. Meanwhile, the concentration of liquid sulfate aerosols used for homogeneous

nucleation in our cirrus sub-model is around at least $10^4 \, L^{-1}$ in the NH, similar to the total aerosol concentration used for heterogeneous nucleation in cirrus (i.e. on dust and soot particles, not shown). For the high-seeding case the large concentration of seeding particles forms numerous ice crystals that consume water vapor and prevent the development of large $S_i$ required for homogeneous nucleation, whereas for the other two cases it appears that the addition of seeding particles merely adds to the ICNC HET. Ice crystals originating from homogeneous nucleation tend to be small due to large competition for water

vapor. Replacing them with heterogeneously-nucleated ice crystals that form at much lower $S_i$ and have a longer time to grow, as in the high-seeding case, means that the overall effect on ice crystal size should perhaps be an increase. However, as the seeding particles themselves are so small (0.01 μm), combined with their high number concentration, it is likely that they





form numerous ice crystals that remain small due to rapid water vapor consumption such that the average ice crystal size remains roughly the same. To investigate this further, Fig. 8 shows the NH wintertime zonal mean cirrus ice sources for the three r0.01 low, mid, and high-seeding cases, including ICNC HOM and the sources terms for ICNC HET for ice forming on mineral dust, soot, and seeding particles. Consistent with the zonal anomaly in Fig. 7c, homogeneous nucleation is almost entirely replaced by the large injection of seeding particles for the high-seeding case (Fig. 8d and Fig. 8p). It also shows that background heterogeneous nucleation on soot and mineral dust in the troposphere are overtaken by nucleation on the high concnetration of seeding particles. At the same time we find additional ice crystals originating from heterogeneous nucleation on background dust particles in the stratosphere. Examining SW and LW heating rates in this region (Appendix C) reveals a weak SW heating rate of up to $0.3\,\mathrm{K/day}$ relative to a stronger LW heating rate of $1.6\,\mathrm{K/day}$ from the optically thicker cirrus in the tropopause region. In turn this induces a strong LW cloud-top cooling effect (Possner et al., 2017; Eirund et al., 2019) of about $-1.2\,\mathrm{K/day}$ above roughly $200\,\mathrm{hPa}$, which coincides with the area of positive ICNC Dust in Fig. 8h. Lower temperatures in the lowermost stratosphere combined with a higher availability of water vapor in this region, as denoted by a positive specific humidity anomaly around $5\,\mathrm{mg/kg}$ on average, forms the conditions for ice nucleation on the background mineral dust particles (up to $100\,\mathrm{L}^{-1}$). As the availability of water vapor in this region is sparse relative to the lower-lying atmosphere, combined with the influx of numerous seeding INPs, the ice crystal growth is limited. Thus, in the high-seeding case, the ice crystal size anomaly we find is the result of limited ice growth on the numerous injected seeding particles or on enhanced background mineral dust INP activity, which produces ice that is comparable in size to the reference unseeded case. The small indication of larger ice crystals in Fig. 7b may be in line with some enhanced sedimentation in the high-seeding case, as noted by the positive ICNC anomaly for heterogeneous nucleation on seeding particles at low altitudes in Fig. 8p.

It remains unclear why the two cases with lower concentrations of seeding particles (low and mid-seeding) produce larger reductions in the average size of ice crystals than the high-seeding case (Fig. 7b). The zonal mean cirrus ice source anomalies in Fig. 8 (second and third columns) reveal an interesting effect on ice nucleation competition. We still find a large positive seeding signal up to about $1000\,\mathrm{L}^{-1}$ for the low-seeding case and exceeding $1000\,\mathrm{L}^{-1}$ for the mid-seeding case. There are also noticeable reductions in both homogeneous and heterogeneous nucleation processes in the troposphere for both cases. However, unlike the high-seeding case, we find a positive homogeneous nucleation anomalies in the lower stratosphere. We also find positive anomalies for heterogeneous nucleation on mineral dust in this region. Thus, it appears that seeding in these two cases has two different effects that explain the large negative ice crystal radius anomalies we showed in Fig. 7d. In the troposphere, seeding replaces some background heterogeneous processes. Plus, some of the ice crystals formed on the seeding particles grow quickly and sediment (as indicated by the small positive ICNC Seed anomaly at low levels in Fig. 8n and o) such that the remaining ice crystal size decreases. As a result, we also find that optically thicker cirrus in these two cases form and exert a stronger LW CRE that warmed the upper troposphere, which for the mid-seeding case exceeded $0.1\,\mathrm{K/day}$ (not shown). This also induces a slightly stronger LW cloud-top cooling effect of about $0.1\,\mathrm{K/day}$ in the mid-seeding case that cools the lower stratosphere and increases ice supersaturation. However, as the seeding particle concentrations in these two cases are not as high as in the high-seeding case, the formation of ice onto these particles is insufficient to prevent higher ice supersaturated conditions from developing that are appropriate for heterogeneous nucleation on mineral dust particles and



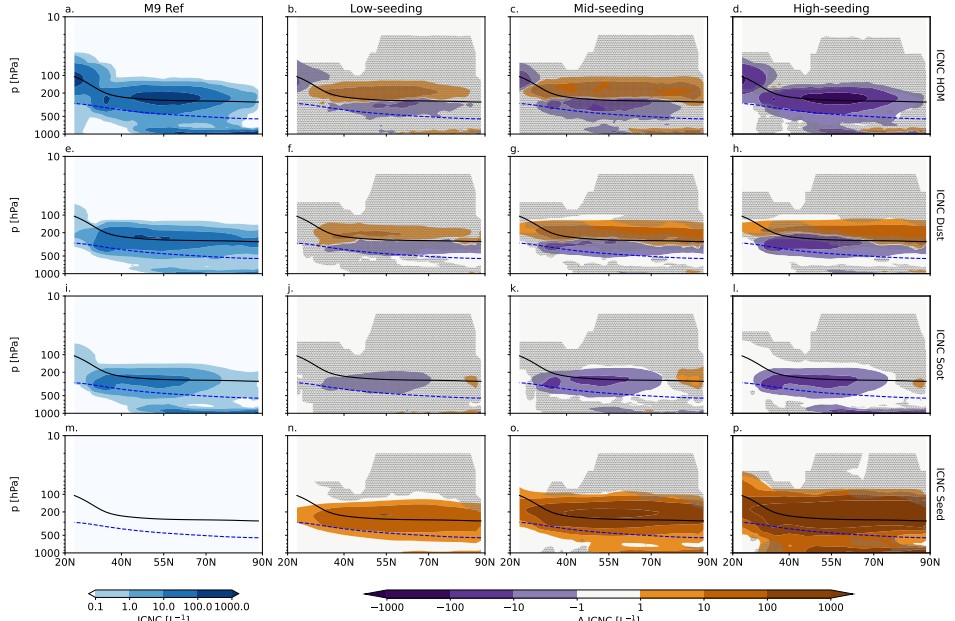

**Figure 8.** Five-year northern hemisphere wintertime zonal mean in-situ cirrus ice sources in $L^{-1}$ for homogeneous nucleation and the sources of heterogeneous nucleation for the unseeded reference case in the first column, and the anomalies for seeding with a factor of 10 (low-seeding, second column), 100 (mid-seeding, third column), and 1000 (high-seeding, fourth column). The first row shows the homogeneous nucleation source. The second, third, and fourth rows each represent a heterogeneous nucleation source for mineral dust (second row), soot (third row), and seeding particles (fourth row). The black line denotes the WMO-defined tropopause and the blue dashed line is the 238 K temperature contour. The stippling denotes insignificant data points at the 95% confidence level according to the independent t-test controlled by the false discovery rate method.

homogeneous nucleation. Both of these signals show wide uncertainty on the 95 % confidence level, but to a lesser extent for the mid-seeding case. Nevertheless, the limited availability of water vapor in the stratosphere limits ice crystal growth.
Therefore, relative to the unseeded reference case, the new ice crystals forming in the stratosphere in this case are smaller. This behavior also explains the vertical mean ice crystal radius anomalies we found in the global seeding cases in Fig. 5.

It is clear that injecting highly efficient seeding particles in our model has widespread effects on ice nucleation in cirrus, with a clear overseeding response for the largest number concentration of seeding particles. We also find impacts on lower-lying mixed phase clouds similar to those found by Tully et al. (2022a). Fig. 9 presents the vertical IWC and liquid water content
(LWC) anomalies averaged over the NH during Novemver to February for all r0.01 cases with mass emission scaling. We find a large increase in ice mass in the cirrus regime (T < 238 K, Fig. 9a) that scales with the number of injected seeding particles. As shown above, this is the result of more numerous and smaller ice crystals that formed on the injected seeding particles. This appears to have an impact on ice crystal sedimentation, at least for the mid and high-seeding cases, as we find negative IWC anomalies in the lower mixed-phase regime. The very small positive IWC anomaly for the low-seeding case is likely the result



of a small number of ice crystals that formed on seeding particles and grew rapidly and then underwent sedimentation to lower

levels in the mixed-phase regime. The reduction of IWC in the mixed-phase regime for the other cases results in an increase in

the liquid water mass (Fig. 9b) due to less efficient mixed-phase processes such as riming and the WBF process that consume

liquid droplets. For the high-seeding case, this increase in LWC in the mixed-phase regime produces a stronger SW CRE of

roughly $1.4\,\mathrm{Wm}^{-2}$ (i.e. more negative), but this is outweighed by the large increase in the LW CRE in this case (by $3.5\,\mathrm{Wm}^{-2}$)

in response to overseeding the cirrus regime (Tab. 4). The results from seeding in the NH during November to February suggest

that the seeder-feeder mechanism (e.g., Choularton and Perry, 1986; Robichaud and Austin, 1988; Reinking et al., 2000; Roe,

2005) appears to be an important source of ice in the mixed-phase regime in our model. This is in-line with observations of the

seeder-feeder process in orographic clouds (Dore et al., 1999; Borys et al., 2003; Purdy et al., 2005; Proske et al., 2021; Ramelli

et al., 2021). However, the main finding is that cirrus clouds in our model show high sensitivity to large perturbations of small

INPs. Gasparini et al. (2020), who also who used ECHAM-HAM, found a lower sensitivity to seeding INP perturbations. They

assumed particles had radii of $50\,\mu\mathrm{m}$ and prescribed a concentration of $1\,\mathrm{L}^{-1}$. Using such large particles in our version of the

model (with HAM M9, Sec. 2.2) did not produce any significant signal from seeding. In order to achieve any appreciable signal

we had to assume much smaller seeding particles, which based on the number concentration mapping from the mass emissions

led to much larger concentrations of seeding INPs. Therefore, our new approach introduces a particle size bias that enhances

the sensitivity we find from our model to seeding INP perturbations.




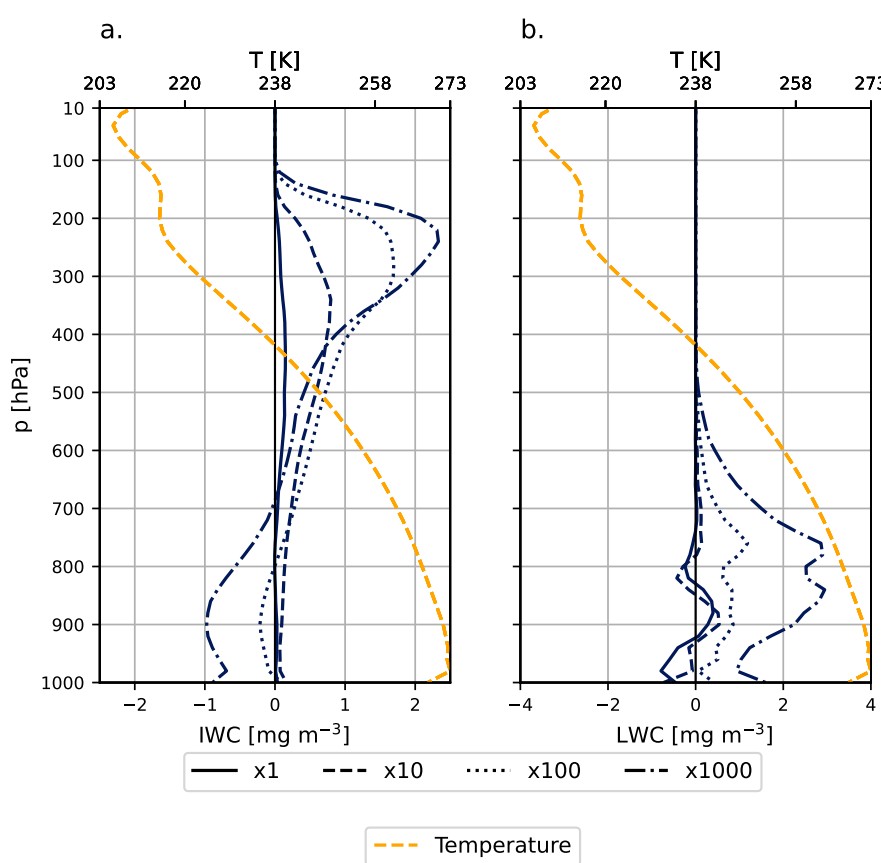

**Figure 9.** Five-year vertical mean anomalies as a function of pressure for (a) IWC and (b) LWC for the NH during the period November to February for seeding with an emissions radius of 0.01 μm for a mass scaling factor of one (solid line), 10 (dashed line), 100 (dotted line), and 1000 (dot-dashed line). The orange dotted line represents the 5-year NH November-February mean vertical profile of temperature centred around the demarcation of the cirrus regime (238 K).





### 3.3 Discussion

The aim of CCT efforts is to create a switch in the dominant ice formation mechanism in cirrus away from homogeneous nucleation and toward heterogeneous nucleation (Storelvmo et al., 2013). Our results show that we achieve this switch; however, it is significant on the 95 % confidence level only for the r0.01 high-seeding case for both global and NH wintertime seeding scenarios. On top of that, the seeding particle number concentration in this case is so high, $> 10^5 \, \mathrm{L}^{-1}$, that the number of ice crystals formed on these particles exceeds the ICNC in the reference unseeded case, originating from both homogeneous and heterogeneous nucleation. In fact, by injecting such a large concentration of seeding particles that can initiate ice formation at a relatively low $S_i$ of 1.05, we find that heterogeneous nucleation processes on background mineral dust and soot INPs are supressed. However, we found that this leads to additional ice formation on background particles in the stratosphere through radiative feedback mechanisms that contribute to the large warming effect we find from overseeding. The overseeding responses are similar to previous CCT findings using ECHAM-HAM, namely Gasparini and Lohmann (2016) and Tully et al. (2022a). However, these past studies used globally uniform distributions of seeding particles with radii of 0.5 µm and number concentration maxima of $100 \, \mathrm{L}^{-1}$. Using the same seeding particle radius following our new prognostic approach resulted in a maximum concentration that just exceeded $10^{-4} \, \mathrm{L}^{-1}$ (Appendix A). The efficacy of seeding with larger particles (radii of 50 µm) was examined by Gasparini et al. (2017) and extended by Gasparini et al. (2020). The earlier study tested several number concentrations of seeding particles, but they found the largest cooling effect of -0.85 $\mathrm{W m}^{-2}$ for a seeding particle concentration of $1 \, \mathrm{L}^{-1}$, which was also found by the latter study. Following our new approach, this size of seeding particle corresponded to a number concentration that was several orders of magnitude smaller ($< 10^{-10} \, \mathrm{L}^{-1}$, Appendix A), and produced negligible effects on the TOA radiative balance as well as on cirrus properties.

By assuming smaller seeding particle radii and by scaling the mass emissions, we found seeding particle concentrations around the same order of magnitude to those used by Tully et al. (2022a). For example, the r0.01 low-seeding and r0.1 high-seeding cases showed maximum seeding particles number concentrations between $100 \, \mathrm{L}^{-1}$ and $1000 \, \mathrm{L}^{-1}$, exceeding the maximum concentration of $100 \, \mathrm{L}^{-1}$ used by Tully et al. (2022a). While for the r0.01 low-seeding case this led to a small warming of about 0.3 $\mathrm{W m}^{-2}$, for the r0.1 high-seeding case this led to a small cooling of 0.02 $\mathrm{W m}^{-2}$. However both of these signals are highly uncertain on the 95 % confidence level (Tab. 3). On the one hand, this means that the chance of producing an overseeding effect by using a more variable spatial and temporal distribution of seeding particles following our new approach is greatly reduced. On the other hand we introduce a bias with this new aircraft seeding approach as we only find a significant signal when we assume the smallest seeding particles (radius = 0.01 µm) with large mass emissions scaling factors (x100 and x1000). This means that it is more likely to produce smaller ice crystals with our new seeding approach, which is exactly what we found for all of the r0.01 cases and some of the r0.1 cases (Fig. 5 and Fig. 7).

One clear finding is that we can also confirm that seeding tropical regions is undesired (Storelvmo and Herger, 2014; Storelvmo et al., 2014; Gasparini et al., 2017). In these regions we find that seeding particles are either ineffective at shutting off homogeneous nucleation or are so effective they over-take this process to produce more ice crystals that were existent in the unperturbed cirrus. When we restrict seeding particle emissions to the NH wintertime, we also only find either warming



from overseeding or small and insignificant effects with lower seeding particle concentrations. This is partly due to background assumptions in our cirrus model pertaining to the role of pre-existing ice crystals. Gasparini et al. (2020) and Tully et al. (2022a) note that the inclusion of vapor deposition on to pre-existing ice crystals makes CCT less effective than models that did not include this process, namely Storelvmo et al. (2013), Storelvmo and Herger (2014), and Storelvmo et al. (2014). Finally, our results also call into question the reliability of CCT to act as a CI strategy. Specifically, the proposed delivery method of seeding

material via commercial aircraft is uncertain as based on our results this introduces a particle size bias in order to achieve a significant signal.

## 4    Conclusions

In this study, we made the first attempt in a GCM to simulate CCT using a fully prognostic aerosol species specifically for cirrus seeding particles. We achieved this by extending the seven-mode aerosol microphysics model, HAM (Stier et al., 2005;

Zhang et al., 2012; Tegen et al., 2019), to include two extra modes to simulate the atmospheric evolution of internally and externally mixed seeding particles made of bismuth-triiodide (Mitchell and Finnegan, 2009). Seeding particle emissions were assumed to follow aircraft emissions of black carbon (soot) particles, following the proposed real-world delivery mechanism (Mitchell and Finnegan, 2009). We found that compared to assuming a globally uniform seeding particle distribution, using aircraft emissions drastically reduces the number concentration of seeding particles available as INPs in our cirrus sub-model.

However, this requires using much smaller seeding particles with high mass emissions scaling in order to achieve a significant signal, which we found always led to overseeding and associated warming.

Aerosol-ice-cloud interactions remain one of the largest uncertainties in the understanding of the climate system (Storelvmo, 2017; Bellouin et al., 2020). This knowledge gap impacts the ability to assess the efficacy CCT to act as a feasible CI strategy, with widely different responses from different models. There are a number of lines of work that could be addressed in this

regard. First, while new evidence suggests that mineral dust is the most prevalent INP species in cirrus globally, with peaks downstream of large source regions (Froyd et al., 2022), it is unclear what role this plays in remote regions like the Arctic. Second, emitting seeding particles from commercial aircraft was proposed as a potential delivery mechanism in the real-world (Mitchell and Finnegan, 2009). However, aircraft emissions of soot contribute an uncertain effect on cirrus, mostly from uncertainty surrounding the ability of soot to act as an INP (Mahrt et al., 2018, 2020; Lee et al., 2021). Finally, our results

also showed that INP perturbations in the cirrus regime ($T < 238\,\mathrm{K}$) had effects on clouds in warmer temperature regimes, namely mixed-phase clouds, which highlighted the importance of the seeder-feeder mechanism in our model. Further work investigating this mechanism under unperturbed scenarios (i.e. without INP injections for CCT) in our model would be useful to understand its importance on mixed-phase processes, including precipitation formation. Overall, however, with such high uncertainty surrounding INP perturbation effects on cirrus, we recommend that more observational evidence is needed on cirrus

formation mechanisms and the impact that natural as well as anthropogenic aerosol have on cirrus properties before further modeling studies proceed with assessing CCT.



## Appendix A: Initial tests of prognostic seeding following aircraft emissions

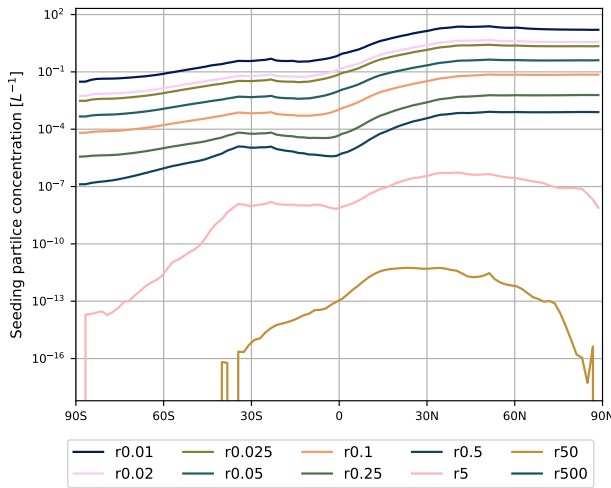

**Figure A1.** Five-year zonal mean seeding particle concentrations (in $L^{-1}$) for the ten initial tests conducted for prognostic seeding following aircraft emissions. Each color represents one of the seeding particle radii that we tested: 0.01, 0.02, 0.025, 0.05, 0.1, 0.25, 0.5, 1, 5, 50, and 500 μm. Note that the number concentration of seeding particles for the case with a radius of 500 μm was so low that it does not show on this scale.

## Appendix B: Heterogeneous nucleation sources - global aircraft seeding

Fig. B1 presents the five-year annual zonal mean cirrus ice source anomalies for the sources of heterogeneous nucleation for the extreme r0.01 high-seeding case as described in the main text. The sources of heterogeneous nucleation include background mineral dust (ICNC Dust) and soot (ICNC Soot) particles, and on cirrus seeding particles (ICNC Seed). In this extreme scenario ice nucleation onto the high concentration of seeding particles not only overtakes homogeneous nucleation, but also on background INPs as shown by the negative anomalies in Fig. B1b and B1d. As described in the main text, outside of the tropics this large increase in the number of new ice crystals in the lowermost stratosphere (above the black line in Fig. B1f) releases latent heat that warms this region and prevents cold temperatures required for homogeneous nucleation. However, this allows heterogeneous nucleation on the small number of background mineral dust particles to also nucleate in this region. Thus, we find positive ICNC Dust anomalies in the lowermost stratosphere outside of the tropics by up to $100\,L^{-1}$ (Fig. B1b).





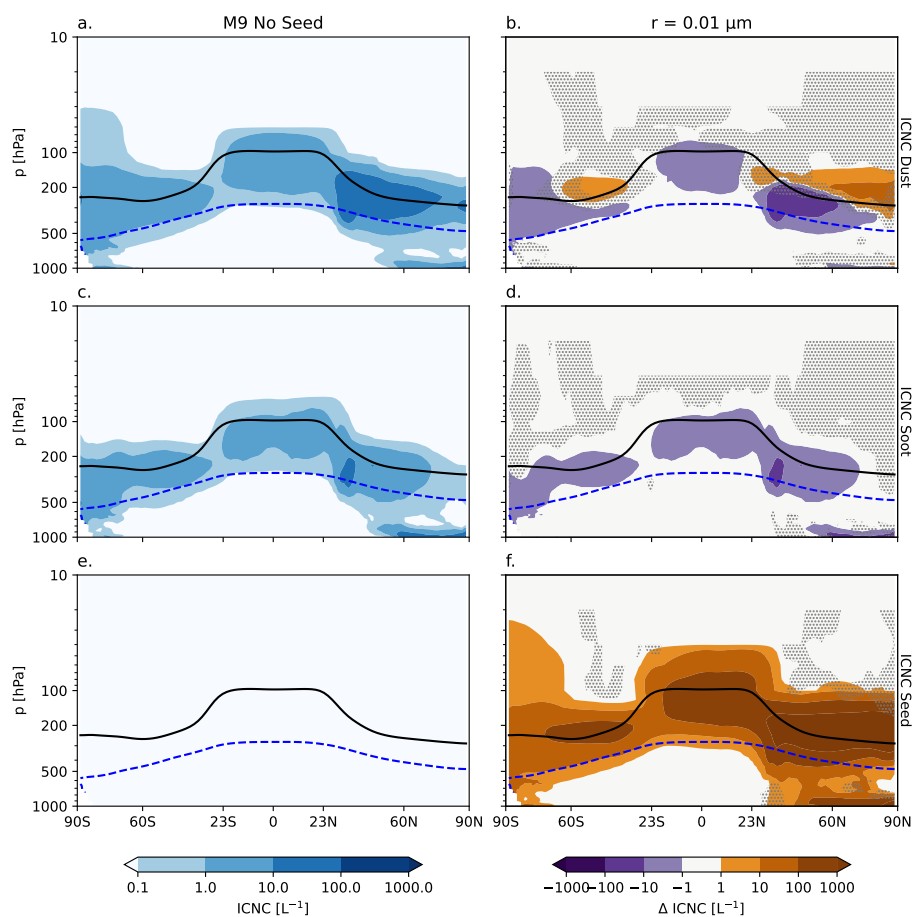

**Figure B1.** Five-year annual zonal mean in-situ cirrus heterogeneous nucleation ice sources in $\mathrm{L}^{-1}$ for the unseeded reference case in the first column, and the anomalies for the seeding scenario r0.01 high-seeding (emissions radius of $0.01\,\mu\mathrm{m}$ and a mass emissions scaling factor of 1000) in the second column. Each row represents a heterogeneous nucleation source for mineral dust (first row), soot (second row), and seeding particles (third row). The black line denotes the WMO-defined tropopause and the blue dashed line is the 238 K temperature contour. The stippling denotes insignificant data points at the 95% confidence level according to the independent t-test controlled by the false discovery rate method.



# Appendix C: Northern hemisphere wintertime TOA radiative anomalies and zonal heating rates

**Table C1.** Five-year NH mean net TOA and net CRE radiative balance anomalies in $\mathrm{Wm^{-2}}$, as well as their SW and LW components for the period between November and February for each of the seeding particle emission radii tested for the NH wintertime seeding scenario. Each quantity includes the 95 % confidence interval equating to 2 standard deviations of the mean values of the 5-year data set. Values in bold denote those that are statistically distinct from zero based on the 95 % confidence level.

| Seeding particle emission radius | net TOA | TOA SW | TOA LW | net CRE | SWCRE | LWCRE |
|---|---|---|---|---|---|---|
| μm | | | x1 | | | |
| 0.01 | 0.17 ± 0.63 | -0.11 ± 0.51 | 0.27 ± 0.61 | 0.33 ± 0.67 | 0.00 ± 0.46 | 0.33 ± 0.38 |
| 0.1 | -0.02 ± 0.64 | 0.00 ± 0.49 | -0.02 ± 0.63 | -0.02 ± 0.66 | 0.00 ± 0.45 | -0.02 ± 0.35 |
| 1 | -0.02 ± 0.62 | 0.03 ± 0.51 | -0.05 ± 0.62 | -0.02 ± 0.66 | 0.02 ± 0.45 | -0.04 ± 0.35 |
| | | | x10 | | | |
| 0.01 | **2.28 ± 0.67** | **-0.86 ± 0.53** | **3.14 ± 0.69** | **2.63 ± 0.70** | **-0.67 ± 0.45** | **3.29 ± 0.48** |
| 0.1 | 0.01 ± 0.61 | -0.01 ± 0.50 | 0.02 ± 0.62 | 0.03 ± 0.65 | 0.00 ± 0.44 | 0.03 ± 0.35 |
| 1 | 0.00 ± 0.62 | -0.01 ± 0.51 | 0.01 ± 0.62 | -0.01 ± 0.67 | -0.01 ± 0.46 | 0.00 ± 0.35 |
| | | | x100 | | | |
| 0.01 | **9.65 ± 0.77** | **-4.15 ± 0.61** | **13.80 ± 0.87** | **9.96 ± 0.85** | **-3.97 ± 0.46** | **13.93 ± 0.77** |
| 0.1 | -0.02 ± 0.64 | 0.00 ± 0.49 | -0.02 ± 0.63 | -0.02 ± 0.66 | 0.00 ± 0.45 | -0.02 ± 0.35 |
| 1 | -0.02 ± 0.62 | 0.03 ± 0.51 | -0.05 ± 0.62 | -0.02 ± 0.66 | 0.02 ± 0.45 | -0.04 ± 0.35 |
| | | | x1000 | | | |
| 0.01 | **12.74 ± 0.90** | **-7.27 ± 0.57** | **20.01 ± 0.95** | **13.38 ± 0.91** | **-7.27 ± 0.47** | **20.65 ± 0.95** |
| 0.1 | 0.59 ± 0.62 | -0.30 ± 0.50 | **0.89 ± 0.61** | **0.81 ± 0.66** | -0.22 ± 0.45 | **1.03 ± 0.37** |
| 1 | 0.02 ± 0.63 | -0.02 ± 0.49 | 0.04 ± 0.64 | 0.03 ± 0.67 | -0.01 ± 0.45 | 0.04 ± 0.35 |





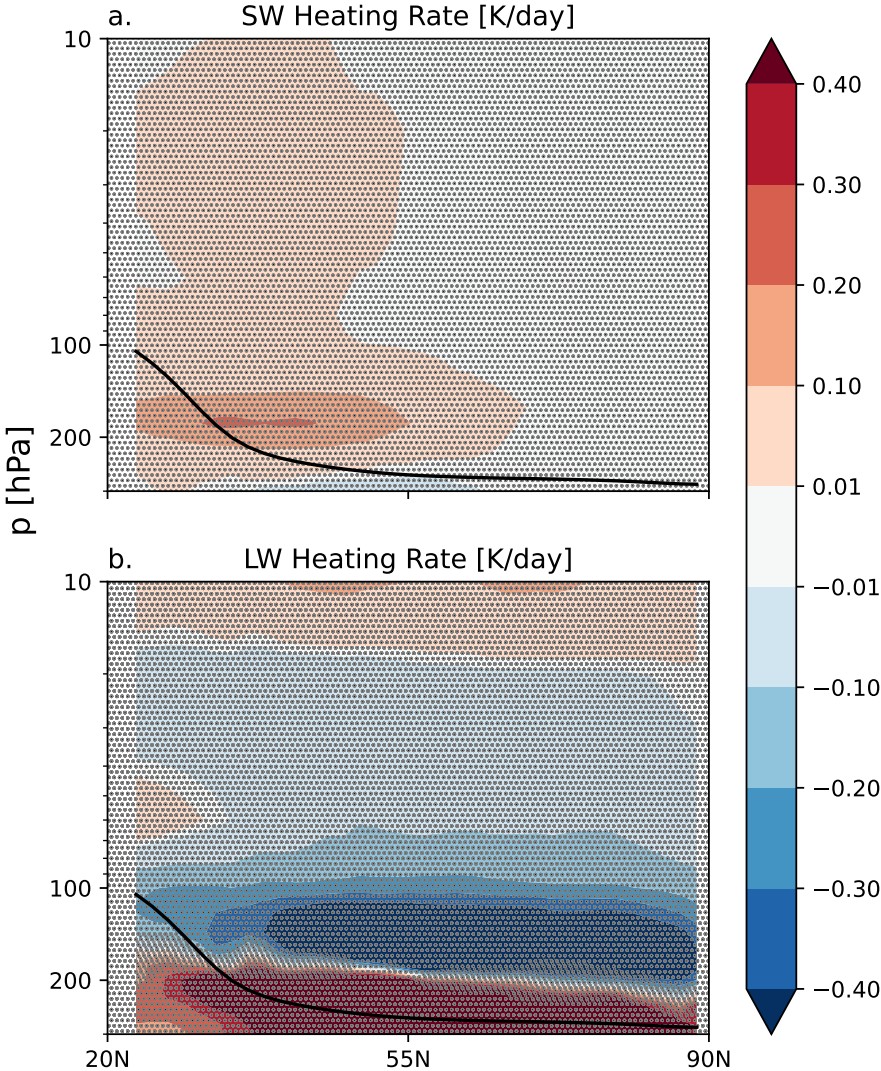

**Figure C1.** Five-year northern hemisphere zonal mean SW and LW heating rate anomalies in $\mathrm{K/day}$ for the r0.01 high-seeding case (emissions radius of $0.01\,\mu\mathrm{m}$ and a mass emissions scaling factor of 1000) for the period November to February. The black line denotes the WMO-defined tropopause and the blue dashed line is the $238\,\mathrm{K}$ temperature contour. The stippling denotes insignificant data points at the 95% confidence level according to the independent t-test controlled by the false discovery rate method.



*Code and data availability.* The ECHAM-HAMMOZ model is freely available to the scientific community under the HAMMOZ Soft-
ware License Agreement, which defines the conditions under which the model can be used (https://redmine.hammoz.ethz.ch/projects/
hammoz/wiki/2_How_to_get_the_sources, last access: 08 November 2022). The version of the code used for this study is archived in
the ECHAM-HAMMOZ SVN repository at https://svn.iac.ethz.ch/external/echam-hammoz/echam6-hammoz/tags/papers/2022/Tully_et_
al_2022_ACPD_M9 (last access: 08 November 2022). Additional information about the model can be found on the HAMMOZ website
(https://redmine.hammoz.ethz.ch/projects/hammoz, (last access: 08 November 2022). The box model that is based on the ECHAM-HAM
code that was used to produce the heterogeneous nucleation-only plots in this manuscript, as well as other post-processing and analysis
scripts are archived on Zenodo (Tully et al., 2022c). The processed GCM output data to produce the relevant plots in this manuscript are also
available on Zenodo (Tully et al., 2022b)

*Author contributions.* CT extended the aerosol microphysics model, HAM, to include two prognostic aerosol species for seeding particles.
CT, DN, and DV worked together to verify this approach and update the in-cloud scavenging procedure in the wet deposition scheme. CT
and UL designed the experiments and CT ran the model simulations, analysed the data, including running the post-processing and plotting
scripts, and wrote the manuscript with comments from all co-authors. UL and DV helped with the interpretation of the results.

*Competing interests.* The authors declare that they have no conflict of interest.

*Acknowledgements.* This Project is funded by the European Union under the Grant Agreement No. 875036 (ACACIA). This work was also
supported by a grant from the Swiss National Supercomputing Centre (CSCS) under project ID s1144.



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
