# Peer review of "Does prognostic seeding along flight tracks produce the desired effects of cirrus cloud thinning?"

_EGUsphere, 2022_

## Referee Comment (RC1)

Review of

**Does prognostic seeding along flight tracks produce the desired effects of cirrus cloud thinning?**

by
Colin Tully , David Neubauer , Diego Villanueva and Ulrike Lohmann

**General:**

In this study, the climate impact of cirrus cloud thinning due to artificially induced ice nucleating particles (INPs) is assessed by using a new approach to simulate the injection of the INPs into the atmosphere. In view of the ongoing anthropogenic climate warming, the topic is of great current relevance. However, in all earlier cirrus geoengineering studies, INPs have been globally equally distributed in the atmosphere. Here, they are seeded only along aircraft flight tracks, which is a much more realistic approach, as in reality that would be the way to bring the particles into the atmosphere. The results are then compared to the globally uniform approach: seeding along the flight tracks significantly reduces the number of INPs compared to global seeding. Therefore, to achieve a significant signal in the radiative feedback of the cirrus, the properties of the INPs had to be set to unrealistic values in terms of size and concentration (very small particles with very high concentrations). However, this always led to overseeding associated with warming, because instead of fewer and larger, more and smaller ice particles formed in comparison to natural conditions. That means, this cirrus geoengineering approach also does not lead to the desired result.

Overall, this is an important study that will help to close the debate on the likelihood of being able to reduce the global warming by geoengineering of cirrus clouds that has been lively discussed over the last decade. For that reason, the manuscript is well within the scope and will be a valuable contribution to ACP. I recommend the study for pulication after considering some comments / questions which are listed below - they are mostly (but not all) minor, intended to improve the Figures and make the study more fluently readable.

**Specific comments:**

**1) Section 3.1**: To my feeling, section 3.1 needs more structure so that the important results are easier to locate. Below I suggest some sub-sections where I think it is helpful.

**2) Line 284ff**: As expected, we find the largest positive net TOA anomaly when seeding with the largest average seeding particle number concentration (> 105 L−1 , Fig. 2) that is associated with the case with a mean emissions radius of 0.01 μm (r0.01) and a mass scaling factor of 1000 (high-seeding).

*Comment:* Why it is expected that a large number of seeding INPs result in a large positive net TOA?

And why does the size matter ? For example, the INP concentration from r0.1 and x1000 is almost identical with r0.01 and x10 /Fig. 4.2), but without a response (Fig. 4.3). Why is that ?
In nature, we know that mostly larger INPS (> 0.5 μm) will be activated, how does that correspond with your finding ?

If it is explained later in the paper what causes the radiation feedbacks, please note that here (cross reference).

**3) Line 286 f**: The large TOA anomalies are driven by a large increase in the LW cloud radiative effect (CRE) by 10.1 Wm−2 (Tab. 3), indicating a significant change in cirrus cloud properties.

*Comment:*  Please specify 'properties' (see also comment to Table 3).

**4) Line 321 – the following paragraph:**

*Comment:* This paragraph could be a sub-section with the title ‚Ice crystal sizes‘

*Question on the paragraph:* Why you discuss here in detail the size anomaly? Is this because the size is one parameter influencing the radiative feedback ?
Because less sedimenation of smaller ice particles keeps the cirrus at higher altitudes (--> more warming) ?    Please explain.

**5) Line 336 – the following paragraph:**

*Comment:* This paragraph could be a sub-section with the title ‚Tropics‘.

 **6) Line 340**  …. by up to -10 mg m−3 (Figure 4.4h).   (*Comment*: line break here)
  Nevertheless, the main effect we find in the tropics is the formation of a large number ...

**7) Line 350 – 374:**  *Comment:* This  could be a sub-section with the title ‚Northern Hemisphere‘.

 **8) Line 352:**  ...we find positive ICNC HET anomalies up to 1000 L−1 at lower levels and a reduction of IWC up to 1.0 mg.m−3  (see Figure 4, f,h).

 **9) Line 359f:** This directly influences the large positive LW CRE we find for the r0.01 high-seeding  case (Fig. 3 and Tab. 3)   *Question*: maybe better Figure 4 b and Figure 6?

**10) Line 375 – end of Section:**  *Comment:* This  could be a sub-section with the title ‚Conclusions: global aircraft seeding‘.

**11) Line 394ff:** However, we restricted seeding particle emissions further by only seeding during NH wintertime (November-February) as this was suggested to optimize cirrus seeding efficacy (Storelvmo and Herger, 2014; Storelvmo et al., 2014).        *Comment:* Please briefly mention why.

**12) Line 435:**  However, as the seeding particles themselves are so small (0.01 µm), combined with their high number concentration, it is likely that they form numerous ice crystals that remain small due to rapid water vapor consumption such that the average ice crystal size remains roughly the same.

*Comment***:** How realistic is the assumption of r = 0.01 µm given that the consequences for CCT are strong but in nature only INP > ~0.5 um form ice crystals ?
→ I think this point should be discussed in the paper in some detail.

**13) Line 441:** At the same time we find higher rates of heterogeneous nucleation on background dust particles in the stratosphere.

*Comment / question***:** Here and at other places of the manuscript:

Wouldn't it be better to call the region above the mean tropopause 'upper tropopause' instead of stratosphere ?
If cirrus clouds form there, then obviously there is enough moisture present - but the 'real' stratosphere is dry so that no cirrus clouds can form.

Another possibility is that the tropopause height increases in comparison to the WMO tropopause height due to the induced warming (Fig. 6) ? Then, what you called stratosphere could be still upper troposphere ?      It could be interesting for the reader to discuss that.

**14)  Line 473f:** Therefore, relative to the unseeded reference case, the new ice crystals forming in the stratosphere in this case are smaller. This behavior also explains the vertical mean ice crystal radius anomalies we found in the global seeding cases in Fig. 5.

*Comment:* It would be good to have a Figure here same as Fig. 4.5, I think seeing the vertical structure if the ice radius anomaly would help understanding the complex processes.

**15) Line 474 – end of Subsection:**  *Comment:* This  could be a sub-section with the title
                                ‚Conclusions: Northern hemisphere-only wintertime seeding'

**16) Line 476f:** Fig. 9 presents the vertical IWC and liquid water content (LWC) anomalies averaged over the NH during Novemver to February for all r0.01 cases with mass emission scaling.

*Comment / Question:* It can be seen from Fig. 9 that the positive IWC anomaly is  mostly below the mean NH tropopause (~250 hPa), though also above numerous ice particles are injected.

From this one can derive that the ice particles above the  WMO tropopause are much smaller so that they do not  cause an IWC anomaly, right?

**17) Line 480f:** As shown above, this is the result of more numerous and smaller ice crystals that formed on the injected seeding particles. This appears to have an impact on ice crystal sedimentation, …

*Comment:* reduced sedimentation because the ice particles are small  - I would mention this instead of stating  imprecisely 'impact'.

**Comments on Figures/Tables:**

**Figure 2:** *Comment:* ‚three emissions radii: 0.01 µm, 0.1 µm, and 1 µm' -  0.01 µm is very small for INP, are they really  activated ??

[Figure]

**Figure 3:  Caption:**  Five-year annual global mean net TOA radiative anomalies (in Wm−2 ) for  each seeding particle emissions mass scaling factor ….

**Figure 4**: *Comment:*  Please note in the title of the right column that this is the case of high seeding; also, please define the solid and the dashed lines

**Figure 5**:

[Figure]

**Figure 6:** *Comment:* Please note in the title of the figure that this is the case of high seeding;

**Figure 7, Caption**: *Comment:* (b) does not show temperature, but Delta_Rice.
**Figure**: *Comment:* please note above or below the figure that the panels are for r = 0.01µm

**Figure 8:**

[Figure]

**Figure 9:** **Figure:** the x-axes, aren't they Delta_LWC and Delta_IWC (not LWC, IWC) ?

Also, please note above or below the figure that the panels are for r = 0.01µm.

**Caption**: Five-year vertical mean anomalies as a function of pressure for (a) IWC and (b) LWC for the NH during the period November to February for seeding with an emissions radius of 0.01 µm for a mass scaling factor of one (solid line), 10 (dashed line), 100 (dotted line),  and 1000 (dot-dashed line).  (a) IWC and (b) LWC: Tthe orange dotted line represents the 5-year NH November-February mean temperature vertical profile centred around the homogeneous freezing temperature limit (238 K).

**Table 1**: *Comment:* You might add to the column 'Freezing method' if all or only
a part of the INPs are activated (AF = 1 or AF = f(x);
AF: activated fraction; x: Si, T, ...)

**Table 3**: *Comment:* please define the solid and the dashed lines

| Seeding particle emission radius | net TOA | TOA SW | TOA LW | net CRE | SWCRE | LWCRE |
|---|---|---|---|---|---|---|
| µm | | | No scaling | please include the scaling factor | | |
| 0.01 | 0.00 ± 0.91 | -0.04 ± 0.61 | 0.04 ± 0.34 | 0.13 ± 0.78 | 0.08 ± 0.81 | 0.05 ± 0.14 |
| 0.1 | 0.00 ± 0.91 | 0.01 ± 0.62 | -0.01 ± 0.34 | 0.00 ± 0.78 | 0.02 ± 0.81 | -0.01 ± 0.13 |
| 1 | 0.02 ± 0.91 | 0.03 ± 0.61 | -0.01 ± 0.34 | 0.02 ± 0.78 | 0.02 ± 0.80 | -0.01 ± 0.13 |
| | | | Low-seeding | | | |
| 0.01 | 0.31 ± 0.91 | -0.37 ± 0.61 | **0.68 ± 0.34** | 0.60 ± 0.77 | -0.18 ± 0.81 | **0.78 ± 0.14** |
| 0.1 | -0.02 ± 0.92 | -0.01 ± 0.61 | -0.01 ± 0.34 | 0.00 ± 0.79 | 0.01 ± 0.81 | -0.01 ± 0.13 |
| 1 | 0.00 ± 0.91 | 0.01 ± 0.61 | 0.00 ± 0.34 | 0.00 ± 0.79 | 0.01 ± 0.81 | -0.01 ± 0.14 |
| | | | Mid-seeding | | | |
| 0.01 | **2.46 ± 0.90** | **-2.34 ± 0.58** | **4.80 ± 0.36** | **2.57 ± 0.77** | **-2.22 ± 0.76** | **4.79 ± 0.13** |
| 0.1 | -0.05 ± 0.90 | -0.04 ± 0.61 | -0.01 ± 0.34 | 0.03 ± 0.78 | 0.01 ± 0.81 | 0.02 ± 0.13 |
| 1 | 0.00 ± 0.91 | 0.01 ± 0.61 | -0.02 ± 0.34 | 0.00 ± 0.78 | 0.01 ± 0.80 | -0.01 ± 0.13 |
| | | | High-seeding | | | |
| 0.01 | **5.94 ± 0.86** | **-5.05 ± 0.56** | **10.99 ± 0.36** | **5.04 ± 0.72** | **-5.06 ± 0.75** | **10.10 ± 0.17** |
| 0.1 | -0.02 ± 0.90 | -0.26 ± 0.60 | 0.24 ± 0.34 | 0.19 ± 0.78 | -0.17 ± 0.81 | 0.36 ± 0.13 |
| 1 | -0.01 ± 0.91 | 0.01 ± 0.62 | -0.03 ± 0.33 | -0.01 ± 0.78 | 0.01 ± 0.81 | -0.02 ± 0.13 |

I would suggest to include the IWC / cloud fraction (or what is the driving varialbe?) in the Table, which might give a hint on the reason for the different radiative feedback?

**Table 4**: *Comment*: See comments on Table 3.

---

## Referee Comment (RC2)

ACP review by David Mitchell

Manuscript title: Does prognostic seeding along flight tracks produce the desired effects of cirrus cloud thinning?
Author(s): Colin Tully et al.
MS No.: egusphere-2022-1238
MS type: Research article

General Comments:

This paper is very well written and organized, and the Introduction is particularly well done. Within the context of global climate modeling, there is a lot of interesting analysis, but whether it illuminates the behavior of real cirrus clouds remains in doubt. As stated at the end of Conclusions: "Overall, however, with such high uncertainty surrounding INP perturbation effects on cirrus, we recommend that more observational evidence is needed on cirrus formation mechanisms and the impact that natural as well as anthropogenic aerosol have on cirrus properties before further modeling studies proceed with assessing CCT."

As stated at the end of "Discussion", some of this uncertainty "is partly due to background assumptions in our cirrus model pertaining to the role of pre-existing ice crystals" which makes CCT less effective. I completely agree and would like to draw the authors attention to a recent ACPD paper by Dekoutsidis et al. (2022). This study evaluates lidar-based water vapor measurements made during the ML Cirrus airborne campaign and describes the distribution and temporal evolution of RHi in cirrus clouds. A key finding was that "The uppermost parts of the clouds are mostly supersaturated with RHi frequently above 140%. That is where new ice crystals form", and where RHi is "reaching the threshold for homogeneous nucleation". That is, homogeneous ice nucleation or hom is likely occurring in a relatively thin layer near cloud top and seems to occur only during the "mature" stage of the cloud. Thus, aircraft measurements are likely to miss these hom events both spatially and temporally. Moreover, spiral descents by aircraft through cirrus (e.g., Mitchell, JAS, 1994) show IWC near cloud top ~ 1/10[th] the IWC near cloud base, suggesting the pre-existing ice assumption may be flawed if it invokes the model layer mean IWC. A typical cirrus cloud might be ~ 1.5 km thick, comparable with a model layer in the UT. The pre-existing ice treatment described in Shi et al. (2015, ACP) is based on the supersaturation development equation that can be written as:

$$\frac{dS_i}{dt} = a_1 S_i W - (a_2 + a_3 S_i) \left( \frac{dq_{i,nuc}}{dt} + \frac{dq_{i,pre}}{dt} \right)$$

where $q_{i,nuc}$ is the ice mass mixing ratio due to nucleation and $q_{i,pre}$ is the ice mass mixing ratio of pre-existing ice, parameters $a_1$, $a_2$, and $a_3$ depend only on the ambient temperature and pressure, $S_i$ is the supersaturation with respect to ice, W is the updraft velocity and t is time.

From this equation it is seen that the greater $q_{i,pre}$ is, the smaller the increase in $S_i$ is. This study by Dekoutsidis et al. implies that $q_{i,pre}$ may be overestimated in GCMs since $q_{i,pre}$ is based on layer mean IWC or q values, whereas the actual $q_{i,pre}$ should correspond to a thin layer near cloud top (where $q_{i,pre} < q_{i,mean}$) that model vertical resolution cannot accommodate. The study by Diao et al. (2015, JGR) shows that ice nucleation in cirrus occurs near cloud top. The modeling results of Spichtinger and Geirens (2009, ACP) appear consistent with these considerations, showing ice crystal production near cloud top and crystal growth at lower levels, which lowers RHi and quenches hom.

For this reason, I question the results in this study and agree with the authors that "more observational evidence is needed on cirrus formation mechanisms". That is, an inflated $q_{i,pre}$ will depress RHi and generally prevent the RHi from reaching the threshold for hom, forcing heterogeneous ice nucleation to occur much more than it otherwise would. According to Shi et al. (2015), "The pre-existing ice crystals significantly reduce ice number concentrations in cirrus clouds, especially at mid- to high latitudes in the upper troposphere (by a factor of ~ 10). Furthermore, the contribution of heterogeneous ice nucleation to cirrus ice crystal number increases considerably." The authors do a good job of mentioning how the pre-existing ice treatment promotes het, but they can also mention the limitations noted above.

Since hom is sensitive to the cooling rate that is determined by the cloud updraft, the treatment of cloud updrafts is critical. The updraft in this ECHAM GCM can be resolved into three components: large scale lifting, TKE turbulence and lifting by orographic gravity waves. Please discuss the treatment of vertical motions in this model and inform the readers whether orographic gravity wave effects were included. These can have a strong impact on cirrus cloud properties (Joos et al., 2008, JGR; Joos et al., 2014, ACP).

The treatment of pre-existing ice appears to assure the dominance of het which would assure that no cooling from seeding occurs, and that CRE changes must be positive. Therefore, any seeding effect will be a warming effect, as shown in Fig. 3. Nonetheless, this study has value in demonstrating the sensitivity of cirrus properties to seeding, regardless of whether CRE is positive or negative. And it demonstrates the limitations of aircraft seeding. However, in regard to aircraft seeding, it could be mentioned that commercial cloud seeding programs produce AgI seeding aerosol mean diameters on the order of 0.01 μm. Mentioning this would make the r0.01 seeding scenarios appear more realistic.

Major comments:

Line 275: Please explain the difference between "global mean net top-of-atmosphere (TOA) and net cloud radiative effect (CRE) anomalies". The former accounts for everything, including RH changes, while the latter pertains to clouds only. Many readers may not know this.

Lines 280-282:  The CCT modeling experiment of Gruber et al. (2019, JGR) shows the impact of CCT on lower mixed phase clouds.  Do their results support this speculation?

Lines 452-3:  This appears true for the mid-seeding case but not the low-seeding case.

Line 454:  Should "Fig. 7d" in this sentence be changed to Fig. 7b?

Lines 472-474:  This explanation makes sense based on other studies, but this study shows ice particle size decreases (and presumably fall speeds as well) with decreasing emission scaling (i.e., decreasing INP concentration).  This explanation thus appears to contradict the preceding discussion.

Lines 529-531:  Could the use of drones make CCT more viable in this respect, as suggested in Mitchell et al. (2011, Cirrus clouds and climate engineering: New findings on ice nucleation and theoretical basis.  In: *Planet Earth 2011 - Global Warming Challenges and Opportunities for Policy and Practice*, Prof. Elias Carayannis (Ed.), ISBN 978-953-307-733-8, InTech, Available from HYPERLINK "http://www.intechopen.com/articles/show/title/cirrus-clouds-and-climate-engineering-new-findings-on-ice-nucleation-and-theoretical-basis").  For example, Storelvmo and Herger (2014) describe a high-latitude seeding approach that would require less flight coverage, and even restricting flights to the Polar Regions would likely result in significant cooling based on their methodology.  It seems plausible to increase the density of drone flights in the Polar Regions to address the concerns of this paper.  Please comment on this.

Technical Comments:

Figure 4 caption:  There is no mention of the solid and dashed curves shown in these plots; these curves should be defined.  They appear to represent the tropopause and the 0°C isotherm.

Figure 7 caption:  The y-axis in Fig. 7b appears to indicate microns (change in ice radius) and not temperature as stated in caption.

Line 470:  Novemver => November

---

## Author Comment (AC1)

**Does prognostic seeding along flight tracks produce the desired effects of cirrus cloud thinning? (egusphere-2022-1238)**

*Colin Tully, David Neubauer, Diego Villanueva, and Ulrike Lohmann*
28th March 2023

Referee #1 Author Response

To Referee #1,

Thank you for taking the time to read and review our manuscript.

I quoted each of your comments below with our responses and changes in the text where applicable.

Sincerely,
Colin Tully (on behalf of all co-authors)

**Specific Comments**

1. **Comment:** To my feeling, section 3.1 needs more structure so that the important results are easier to locate. Below I suggest some sub-sections where I think it is helpful.
    a. **Response:** Thank you. We have addressed your specific comments below.

2. **Comment:** Line 284ff: As expected, we find the largest positive net TOA anomaly when seeding with the largest average seeding particle number concentration ($> 10^5$ L$^{-1}$, Fig. 2) that is associated with the case with a mean emissions radius of 0.01 $\mu$m (r0.01) and a mass scaling factor of 1000 (high-seeding). Why is it expected that a large number of seeding INPs result in a large positive net TOA?
    a. **Response:** This specific case that you quote in the comment is the largest number of INPs we simulated in our study. As this represents the largest perturbation to the unseeded reference case it is expected that we find the largest net TOA anomaly. However, that could be either positive or negative. In the revised text, we cut "As expected,".

    And why does the size matter? For example, the INP concentration from r0.1 and x1000 is almost identical with r0.01 and x10 /Fig. 4.2, but without a response (Fig. 4.3). Why is that?

    a. **Response:** Aerosol size matters in our study as it impacts the rate of removal in the atmosphere, i.e. smaller particles will have a longer atmospheric lifetime and therefore their concentrations accumulate over time.

b.  While the two cases you state above do have very similar INP concentrations, the r0.01x10 concentration is slightly higher on the log-scale in Figure 2 towards higher latitudes. We suspect this is why it shows a larger response than r0.1x1000.

In nature, we know that mostly larger INPS (> 0.5 μm) will be activated, how does that correspond with your finding?

a.  **Response:** Our inclusion of smaller particles is justified based on previous work on the ice nucleation ability of AgI. We included some examples of this work in the revised text under the Experimental Setup section.
b.  **Changes in the text:**

*"Including such small seeding particles in our model (0.01 μm) is justified based on previous work on the ice nucleation ability of silve-iodide (AgI). Xue et al. (2013) formulated a parameterization for glaciogenic cloud seeding with AgI in the Weather Research and Forecasting (WRF) model, using a mean particle diameter of 0.04 μm They reported that the model could reasonably produce the physical processes of cloud seeding. Geresdi et al. (2020) also investigated cloud seeding in the WRF model with slightly larger AgI with a mean diameter of 0.05 μm and reported that the model also reasonably reproduced the microphysical properties of real clouds. Marcolli et al. (2016) reviewed lab-based experiments of ice nucleation and showed that AgI particles of 20 nm in diameter had an increasing ice nucleation efficiency towards cirrus temperatures (238 K). Finally, Kanji et al. (2017) presented new evidence of the ice nucleation ability of small particles such as pollen and fungal spores, which challenges arguments that only large particles are suitable INPs."*

If it is explained later in the paper what causes the radiation feedbacks, please note that here (cross reference).

c.  **Response:** Agreed. We added a reference to the discussion section in the text.
d.  **Changes in the text:**

*"To some extent this could be linked to optically thicker cirrus (Krämer et al., 2020) or optically thicker lower-lying mixed-phase or liquid clouds (Twomey, 1959, 1977; Albrecht, 1989), which is discussed further in Section 3.3."*

3.  **Comment:** Line 286 f: The large TOA anomalies are driven by a large increase in the LW cloud radiative effect (CRE) by 10.1 Wm$^{-2}$ (Tab. 3), indicating a significant change in cirrus cloud properties.

Please specify 'properties' (see also comment to Table 3).

a.  **Response:** See response under your **Comment 27** for Table 3. We agree that this could be expanded, but only slightly. Therefore, we

amended this sentence in the revised text to point the reader to Figure 4 that shows that cloud fraction and increased ICNC are linked to the positive TOA anomalies we find.

b. **Changes in the text:**

*"The large TOA anomalies are driven by a large increase in the LW cloud radiative effect (CRE) by 10.1 Wm$^{-2}$ (Tab. 3), indicating a significant change in cirrus cloud properties such as cloud fraction and ICNC (Fig. 4)."*

4. **Comment:** Line 321 – the following paragraph:

This paragraph could be a sub-section with the title ‚ Ice crystal sizes ‘

*Question on the paragraph:* Why you discuss here in detail the size anomaly? Is this because the size is one parameter influencing the radiative feedback? Because less sedimentation of smaller ice particles keeps the cirrus at higher altitudes (--> more warming)? Please explain.

a. **Response:** We agree with the addition of a sub-section to 3.1 and added this into the revised manuscript.

b. **Response to question:** Yes, exactly as you wrote, we discuss ice crystal size in detail here as it is one parameter impacting the radiative feedback through the longer lifetime of ice crystals in the clouds at high altitudes, which contributes to a stronger positive radiative forcing. We added this to the text.

c. **Changes in the text:**

*"Combined with the large increase in ICNC from the shift of homogeneous to heterogeneous nucleation, this indicates a large reduction in the size of ice crystals, which would contribute to positive CRE anomalies through longer cloud lifetimes from reduced ice crystal sedimentation."*

5. **Comment:** Line 336 – the following paragraph:

This paragraph could be a sub-section with the title ‚ Tropics ‘.

a. **Response:** We also agree here and added another sub-section titled "Tropical response"

6. **Comment:** Line 340 .... by up to -10 mg m–3 (Figure 4.4h). (Comment: line break here) Nevertheless, the main effect we find in the tropics is the formation of a large number ...

a. **Response:** We agree, and this was implemented in the revised text.

7. **Comment:** Line 350 – 374: This could be a sub-section with the title ‚ Northern Hemisphere ‘.
   a. **Response:** We agree. We added a sub-section here in the revised manuscript.

8. **Comment:** Line 352: ...we find positive ICNC HET anomalies up to 1000 L⁻¹ at lower levels and a reduction of IWC up to 1.0 mg.m⁻³ (see Figure 4, f,h).
   a. **Response:** This is incorrect wording. We fixed this in the revised text.
   b. **Changes in the text:**

*"Meanwhile, in the NH we find small positive ice crystal radius anomalies in lower levels (p > 600 hPa) for the r0.01 high-seeding case that is consistent with larger ice crystals that sediment more readily. This may be the case in some regions of the NH where we find positive ICNC HET anomalies up to 1000 $L^{-1}$ at lower levels. This is not reflected in the zonal IWC anomalies at lower levels where we find a reduction of IWC up to 1.0 $mg.m^{-3}$ throughout most of the NH except in the Arctic where it is positive by up to 1.0 $mg.m^{-3}$. However, these low-level IWC signals are insignificant as indicated by the stippling in Fig. 4h."*

9. **Comment:** Line 359f: This directly influences the large positive LW CRE we find for the r0.01 high- seeding case (Fig. 3 and Tab. 3) Question: maybe better Figure 4 b and Figure 6?
   a. **Response:** After re-examining the text and the figures, we agree that it is better to reference Figures 4b and 6 in addition to Table 3. This was amended in the revised text.
   b. **Changes in the text:**

*"This directly influences the large positive LW CRE we find for the r0.01 high- seeding case (Tab. 3, Fig. 4b, and Fig. 6)."*

10. **Comment:** Line 375 – end of Section: This could be a sub-section with the title ‚ Conclusions: global aircraft seeding ‘.
    a. **Response:** We agree, but we named the sub-section "Summary of global aircraft seeding".

11. **Comment:** Line 394ff: However, we restricted seeding particle emissions further by only seeding during NH wintertime (November-February) as this was suggested to optimize cirrus seeding efficacy (Storelvmo and Herger, 2014; Storelvmo et al., 2014). Comment: Please briefly mention why.
    a. **Response:** Good point. We added a short explanation in the text
    b. **Changes in the text:**

*"However, we restricted seeding particle emissions further by only seeding during NH wintertime (November-February) as this was suggested to optimize cirrus seeding efficacy due to a lack of incident solar radiation during this time period, meaning cirrus exert only a LW warming effect (Storelvmo and Hereger, 2014; Storelvmo et al., 2014)."*

12. **Comment:** Line 435: However, as the seeding particles themselves are so small (0.01 $\mu$m), combined with their high number concentration, it is likely that they form numerous ice crystals that remain small due to rapid water vapor consumption such that the average ice crystal size remains roughly the same.

    How realistic is the assumption of r = 0.01 $\mu$m (r0.01) and a mass scaling factor of 1000 (high-m given that the consequences for CCT are strong but in nature only INP > ~0.5 um form ice crystals?
    → I think this point should be discussed in the paper in some detail.

    a. **Response:** It is a large exaggeration and demonstrates the rather extreme conditions we must simulate in our model in order to achieve significant results. This is included in our discussion section where we describe that our new approach reduces the likelihood of overseeding when using seeding particle sizes similar to previous studies with ECHAM-HAM, but introduces a size bias to achieve a significant signal. We extended the first paragraph in the dicussions section to address this size issue in more detail.
    b. **Changes in the text:**

*"It is unclear whether using such small seeding particles is a feasible approach to represent a real-world seeding scenario as larger particles are typically favored as INPs in the atmosphere. While recent research suggests that smaller particles may be ice active (e.g. pollen and fungal spores, Kanji et al., 2017), it is also unclear if this also applies to cirrus conditions."*

13. **Comment:** Line 441: At the same time we find higher rates of heterogeneous nucleation on background dust particles in the stratosphere.

    Comment / question: Here and at other places of the manuscript:

    Wouldn't it be better to call the region above the mean tropopause 'upper tropopause' instead of stratosphere?

    If cirrus clouds form there, then obviously there is enough moisture present - but the 'real' stratosphere is dry so that no cirrus clouds can form.

    Another possibility is that the tropopause height increases in comparison to the WMO tropopause height due to the induced warming (Fig. 6)? Then, what you called stratosphere could be still upper troposphere? It could be interesting for the reader to discuss that.

a. **Response:** We did not consider that point. We changed our wording in the revised text when discussing this region to the upper troposphere, lower stratosphere (UTLS). Below we provide how we explain this in the revised text.
b. **Changes in the text:**

*"As the black line on each plot in Fig. 4 represents the annual mean tropopause height over the five years of our simulations as defined by the World Meteorological Organization (WMO), and to account for tropopause height variations, we refer to this region from now on as the upper-troposphere, lower-stratosphere (UTLS)."*

14. **Comment:** Line 473f: Therefore, relative to the unseeded reference case, the new ice crystals forming in the stratosphere in this case are smaller. This behavior also explains the vertical mean ice crystal radius anomalies we found in the global seeding cases in Fig. 5.

It would be good to have a Figure here same as Fig. 4.5, I think seeing the vertical structure if the ice radius anomaly would help understanding the complex processes.

   a. **Response:** This is a good idea, but to not make the main text too repetitive we added a plot for NH wintertime ice crystal anomalies to Appendix C.
   b. **Changes in the text;**

*"Therefore, relative to the unseeded reference case, the new ice crystals forming in the UTLS in this case are smaller. This behavior also explains the vertical mean ice crystal radius anomalies we found in the global seeding cases in Fig. 5. We present a similar vertical ice crystal anomaly plot for our NH wintertime seeding in Appendix C. While the high-seeding case shows the largest reduction in ice crystal radius around 350 hPa it is partially compensated by larger ice particles at lower levels (p > 600 hPa), which is why we find the smallest zonal ice crystal radius anomalies on all levels for this case in Fig. 7b."*

15. **Comment:** Line 474 – end of Subsection: This could be a sub-section with the title , Conclusions: Northern hemisphere-only wintertime seeding '
   a. **Response:** Similar to your **Comment 10**, we agree but named the sub-section "Summary of northern hemisphere-only wintertime seeding

16. **Comment:** Line 476f: Fig. 9 presents the vertical IWC and liquid water content (LWC) anomalies averaged over the NH during Novemver to February for all r0.01 cases with mass emission scaling.

It can be seen from Fig.9 that the positive IWC anomaly is mostly below the mean NH tropopause (~250 hPa), though also above numerous ice particles are injected.

From this one can derive that the ice particles above the WMO tropopause are much smaller so that they do not cause an IWC anomaly, right?

    a. **Response:** We find that the ice crystals above 250 hPa are smaller, though the peak is around 350 hPa. This is partially related to your **Comment 14**, where we added an ice crystal anomaly plot to the Appendix. We argue that it is the formation of new ice crystals that are on average smaller than the reference case that leads to the positive IWC anomaly at this level. We find only a small positive ice crystal radius anomaly for the r0.01 high-seeding case at high levels ($p < 200$ hPa).

17. **Comment:** Line 480f: As shown above, this is the result of more numerous and smaller ice crystals that formed on the injected seeding particles. This appears to have an impact on ice crystal sedimentation, ...

    Reduced sedimentation because the ice particles are small - I would mention this instead of stating imprecisely 'impact'.

    a. **Response:** We agree, and this was revised to be more precise in the text.
    b. **Changes in the text:**

*"As shown above, this is the result of more numerous and smaller ice crystals that formed on the injected seeding particles. This leads to reduced ice crystal sedimentation, at least for the mid and high-seeding cases, as we find negative IWC anomalies in the lower mixed-phase regime."*

**Comments on Figures/Tables**

18. **Figure 2:** three emissions radii: 0.01 $\mu$ m, 0.1 $\mu$ m, and 1 $\mu$ m' - 0.01 $\mu$ m is very small for INP, are they really activated?

[Figure]

a. **Response:** We agree that for consistency with the text that the qualitative scaling factors should be included. This is reflected in the new version of the figure in the manuscript. We also find the idea of the 1000 L⁻¹ line is a good idea. This was also added to the new figure. Regarding your comment on size, please see our response to your **Comment 2**.

19. **Figure 3:** Caption: Five-year annual global mean net TOA radiative anomalies (in Wm–2 ) for  each seeding particle emissions mass scaling factor ....
    a. **Response:** Thank you for finding this typo. It was fixed in the revised manuscript.

20. **Figure 4**: Please note in the title of the right column that this is the case of high seeding; also, please define the solid and the dashed lines
    a. **Response:** We agree this should be clearer. Thank you for pointing out that we hadn't defined the lines in the figure caption. This is fixed in the revised manuscript.

21. **Figure 5**:

[Figure]

a. **Response:** We agree in line with your other comments on our figure. This was revised in the manuscript.

22. **Figure 6:** Please note in the title of the figure that this is the case of high seeding;
    a. **Response:** We agree, and we added this distinction to the title

23. **Figure 7, Caption**: *Comment:* (b) does not show temperature, but Delta_Rice.
    a. **Response:** Thank you for noting that inconsistency. We fixed this in the revised manuscript.

    **Figure Comment:** please note above or below the figure that the panels are for r = 0.01 μm

    b. **Response:** We agree, and this was added to the revised figure.

24. **Figure 8:**

[Figure]

a. **Response:** We disagree with the labels in column a for each ice crystal source as these are included in the titles on the right and explained in the caption. However, the description of these could be clearer. We do agree that the size of the seeding particles could be included as a title. These edits were added to the revised figure.

b. **Changes in the text:**

*"Five-year northern hemisphere wintertime zonal mean ICNC for in-situ sources of cirrus ice in $L^{-1}$ for homogeneous nucleation and the sources of heterogeneous nucleation for the unseeded reference case in the first column, and the anomalies for seeding with a factor of 10 (low-seeding, second column), 100 (mid-seeding, third column), and 1000 (high-seeding, fourth column). The first row shows the ICNC from homogeneous nucleation (ICNC HOM). The second, third, and fourth rows each represent the ICNC from heterogeneous nucleation sources for mineral dust (second row, ICNC Dust), soot (third row, ICNC Soot), and seeding particles (fourth row, ICNC Seed). The black line denotes the WMO-defined tropopause and the blue dashed line is the 238 K temperature contour. The stippling denotes insignificant data points at the 95% confidence level according to the independent t-test controlled by the false discovery rate method."*

25. **Figure 9:** the x-axes, aren't they Delta_LWC and Delta_IWC (not LWC, IWC)? Also, please note above or below the figure that the panels are for r = 0.01 µm.

**Caption**: Five-year vertical mean anomalies as a function of pressure for  for the NH during the period November to February for seeding with an emissions radius of 0.01 µm (r0.01) and a mass scaling factor of 1000 (high-m for a mass scaling factor of one (solid line), 10 (dashed line), 100 (dotted line), and 1000 (dot- dashed line). (a) IWC and (b) LWC: The orange dotted line represents the 5-year NH November- February mean temperature vertical profile centred around the homogeneous freezing temperature limit (238 K).

    a. **Response:** Yes, the IWC and LWC should include "Δ" in front. This was changed in the revised manuscript. We disagree with the proposed change in the caption and instead found a new way to incorporate this more clearly.

    b. **Changes in the text:**

*"Five-year vertical mean IWC (a) and LWC (b) anomalies as a function of pressure for the NH during the period November to February for seeding with an emissions radius of 0.01 µm (r0.01) and a mass scaling factor of 1000 (high-m for a mass scaling factor of one (solid line), 10 (dashed line), 100 (dotted line), and 1000 (dot- dashed line). The orange dotted line represents the 5-year NH November- February mean temperature vertical profile centred around the homogeneous freezing temperature limit (238 K)."*

26. **Table 1**: You might add to the column 'Freezing method' if all or only a part of the INPs are activated (AF = 1 or AF = f(x); AF: activated fraction; x: Si, T, ...)

    a. **Response:** We agree with this point. It is defined in the text, but this is a clearer way to display this information. We added an extra row in this column to specify how the number of particles that can become ice active is calculated. We also changed the table caption.

    b. **Changes in the text:**

*"Adapted from Tully et al. (2022a). A summary of the different aerosol species available for ice nucleation within the in-situ cirrus sub-model. We also present information on the average radius of the particles, the critical ice saturation ratio above which these particles will nucleate ice, the freezing mechanism by which nucleation will occur, and the freezing method within the context of the cirrus scheme following Muench and Lohmann (2020). Under each freezing method we also include the "ice-activity" as a means to define the number of particles in each category that can nucleate ice. Continuous processes are based on the activated fraction (AF) as a function of temperature (T) and $S_i$, whereas for threshold processes all of the available particles (100%) can nucleate ice unless specified otherwise. "Int. mixed" stands for internally mixed (soluble) aerosol species and "Ext. mixed" stands for externally mixed (insoluble) species. Particle types (i.e. aerosol species) denoted in italics are included as additional processes relative to the base version of our model."*

27. **Table 3**: *Comment:* please define the solid and the dashed lines

a. **Response:** We believe this comment is for Figure 4. Definitions for the solid and dashed lines were added to the figure captions. For table 3 we added the scaling factor to each line for clarity. Regarding your second comment, it is the cloud fraction and the increase in ICNC of the cirrus that substantially contributes to the TOA anomalies we find, both of which are reflected in Figure 4 as zonal means. In order to keep this table focused, we therefore disagree with adding an additional column for CF. However, we agree this could be added to the text to make it clear that this is a determining factor and to make reference to Figure 4. Please see the response under your **Comment 3**.

| Seeding particle emission radius | net TOA | TOA SW | TOA LW | net CRE | SWCRE | LWCRE |
|---|---|---|---|---|---|---|
| μm | | | No scaling | please include the scaling factor | | |
| 0.01 | 0.00 ± 0.91 | -0.04 ± 0.61 | 0.04 ± 0.34 | 0.13 ± 0.78 | 0.08 ± 0.81 | 0.05 ± 0.14 |
| 0.1 | 0.00 ± 0.91 | 0.01 ± 0.62 | -0.01 ± 0.34 | 0.00 ± 0.78 | 0.02 ± 0.81 | -0.01 ± 0.13 |
| 1 | 0.02 ± 0.91 | 0.03 ± 0.61 | -0.01 ± 0.34 | 0.02 ± 0.78 | 0.02 ± 0.80 | -0.01 ± 0.13 |
| | | | Low-seeding | | | |
| 0.01 | 0.31 ± 0.91 | -0.37 ± 0.61 | **0.68 ± 0.34** | 0.60 ± 0.77 | -0.18 ± 0.81 | **0.78 ± 0.14** |
| 0.1 | -0.02 ± 0.92 | -0.01 ± 0.61 | -0.01 ± 0.34 | 0.00 ± 0.79 | 0.01 ± 0.81 | -0.01 ± 0.13 |
| 1 | 0.00 ± 0.91 | 0.01 ± 0.61 | 0.00 ± 0.34 | 0.00 ± 0.79 | 0.01 ± 0.81 | -0.01 ± 0.14 |
| | | | Mid-seeding | | | |
| 0.01 | **2.46 ± 0.90** | **-2.34 ± 0.58** | **4.80 ± 0.36** | **2.57 ± 0.77** | **-2.22 ± 0.76** | **4.79 ± 0.13** |
| 0.1 | -0.05 ± 0.90 | -0.04 ± 0.61 | -0.01 ± 0.34 | 0.03 ± 0.78 | 0.01 ± 0.81 | 0.02 ± 0.13 |
| 1 | 0.00 ± 0.91 | 0.01 ± 0.61 | -0.02 ± 0.34 | 0.00 ± 0.78 | 0.01 ± 0.80 | -0.01 ± 0.13 |
| | | | High-seeding | | | |
| 0.01 | **5.94 ± 0.86** | **-5.05 ± 0.56** | **10.99 ± 0.36** | **5.04 ± 0.72** | **-5.06 ± 0.75** | **10.10 ± 0.17** |
| 0.1 | -0.02 ± 0.90 | -0.26 ± 0.60 | 0.24 ± 0.34 | 0.19 ± 0.78 | -0.17 ± 0.81 | 0.36 ± 0.13 |
| 1 | -0.01 ± 0.91 | 0.01 ± 0.62 | -0.03 ± 0.33 | -0.01 ± 0.78 | 0.01 ± 0.81 | -0.02 ± 0.13 |

I would suggest to include the IWC / cloud fraction (or what is the driving varialbe?) in the Table, which might give a hint on the reason for the different radiative feedback?

28. **Table 4:** see comments on Table 3.
   a. **Response:** Please see response under your **Comments 3 & 27**.

---

## Author Comment (AC2)

**Does prognostic seeding along flight tracks produce the desired effects of cirrus cloud thinning? (egusphere-2022-1238)**

*Colin Tully, David Neubauer, Diego Villanueva, and Ulrike Lohmann*
28th March 2023

Referee #2 Author Response

To David Mitchell,

> Thank you for taking the time to read and review our manuscript, and to provide useful feedback on aeras of improvement of our study.

> I quoted each of your comments below with our responses and changes in the text where applicable.

Sincerely,
Colin Tully (on behalf of all co-authors)

**General Comments**

This paper is very well written and organized, and the Introduction is particularly well done. Within the context of global climate modeling, there is a lot of interesting analysis, but whether it illuminates the behavior of real cirrus clouds remains in doubt. As stated at the end of Conclusions: "Overall, however, with such high uncertainty surrounding INP perturbation effects on cirrus, we recommend that more observational evidence is needed on cirrus formation mechanisms and the impact that natural as well as anthropogenic aerosol have on cirrus properties before further modeling studies proceed with assessing CCT."

**Response:** Thank you. We agree that it is unclear whether this is reflective of real cirrus and highlighted this in our discussion and conclusions, as you state.

1. **Comment:** As stated at the end of "Discussion", some of this uncertainty "is partly due to background assumptions in our cirrus model pertaining to the role of pre-existing ice crystals" which makes CCT less effective. I completely agree and would like to draw the authors attention to a recent ACPD paper by Dekoutsidis et al. (2022). This study evaluates lidar-based water vapor measurements made during the ML Cirrus airborne campaign and describes the distribution and temporal evolution of RHi in cirrus clouds. A key finding was that "The uppermost parts of the clouds are mostly supersaturated with RHi frequently above 140%. That is where new ice crystals form", and where RHi is "reaching the threshold for homogeneous nucleation". That is, homogeneous ice nucleation or hom is likely occurring in a relatively thin layer near cloud top and seems to occur only during the "mature" stage of the cloud. Thus, aircraft measurements are likely to miss these hom events both spatially and temporally. Moreover, spiral descents by aircraft through cirrus (e.g., Mitchell, JAS, 1994) show IWC near cloud top ~ 1/10th the IWC near

cloud base, suggesting the pre-existing ice assumption may be flawed if it invokes the model layer mean IWC. A typical cirrus cloud might be ~ 1.5 km thick, comparable with a model layer in the UT. The pre-existing ice treatment described in Shi et al. (2015, ACP) is based on the supersaturation development equation that can be written as:

$$\frac{dS_i}{dt} = a_1 S_i W - (a_2 + a_3 S_i)\left(\frac{dq_{i,nuc}}{dt} + \frac{dq_{i,pre}}{dt}\right)$$

where $q_{i,nuc}$ is the ice mass mixing ratio due to nucleation and $q_{i,pre}$ is the ice mass mixing ratio of pre-existing ice, parameters $a_1$, $a_2$, and $a_3$ depend only on the ambient temperature and pressure, $S_i$ is the supersaturation with respect to ice, W is the updraft velocity and t is time. From this equation it is seen that the greater $q_{i,pre}$ is, the smaller the increase in $S_i$ is. This study by Dekoutsidis et al. implies that $q_{i,pre}$ may be overestimated in GCMs since $q_{i,pre}$ is based on layer mean IWC or q values, whereas the actual $q_{i,pre}$ should correspond to a thin layer near cloud top (where $q_{i,pre} < q_{i,mean}$) that model vertical resolution cannot accommodate. The study by Diao et al. (2015, JGR) shows that ice nucleation in cirrus occurs near cloud top. The modeling results of Spichtinger and Geirens (2009, ACP) appear consistent with these considerations, showing ice crystal production near cloud top and crystal growth at lower levels, which lowers RHi and quenches hom.

For this reason, I question the results in this study and agree with the authors that "more observational evidence is needed on cirrus formation mechanisms". That is, an inflated $q_{i,pre}$ will depress RHi and generally prevent the RHi from reaching the threshold for hom, forcing heterogeneous ice nucleation to occur much more than it otherwise would. According to Shi et al. (2015), "The pre-existing ice crystals significantly reduce ice number concentrations in cirrus clouds, especially at mid- to high latitudes in the upper troposphere (by a factor of ~ 10). Furthermore, the contribution of heterogeneous ice nucleation to cirrus ice crystal number increases considerably." The authors do a good job of mentioning how the pre-existing ice treatment promotes het, but they can also mention the limitations noted above.

    a. **Response:** This is a good point that we did not consider. Based on that study you cite it does appear that pre-existing ice in our model could be over-predicted, thus impacting the efficacy of CCT. However, our model does not have the necessary vertical resolution at cirrus levels to accurately represent the vertical structure of humidity in cirrus. We extended the discussion section to include this point.

    b. **Changes in the text:**

*"This is partly due to background assumptions in our cirrus model pertaining to the role of pre-existing ice crystals. Gasparini et al. (2020) and Tully et al. (2022) note that the inclusion*

*of vapor deposition on to pre-existing ice crystals makes CCT less effective than models that did not include this process (e.g., Storelvmo et al., 2013; Storelvmo and Hereger, 2014; Storelvmo et al., 2014), due to saturation quenching that reduces $S_i$ and prevents homogeneous nucleation from occurring as frequently in the unseeded cirrus. Recent in-situ measurements suggest that the inclusion of pre-existing ice in our model may be over-predicted. Dekoutsidis et al. (2023) analyzed lidar water vapor measurements to assess the in-cloud relative humidity with respect to ice ($RH_i$) in cirrus. They found that $RH_i$ values often reached the homogeneous nucleation limit (140%) near cloud-top, which coincides with the region within a cloud where new ice crystal formation preferentially occurs. After new ice crystals form, they may grow quickly and sediment and not necessarily have a large impact on in-cloud $RH_i$ at cloud top. Our model does not include sufficient vertical resolution (roughly 700 m at cirrus levels, Gasparini et al., 2016) to resolve the vertical humidity structure in cirrus. This represents a motivation for future work that could aid in resolving the role of pre-existing ice in cirrus, which would have large implications on the efficacy of CCT."*

2. **Comment:** Since hom is sensitive to the cooling rate that is determined by the cloud updraft, the treatment of cloud updrafts is critical. The updraft in this ECHAM GCM can be resolved into three components: large scale lifting, TKE turbulence and lifting by orographic gravity waves. Please discuss the treatment of vertical motions in this model and inform the readers whether orographic gravity wave effects were included. These can have a strong impact on cirrus cloud properties (Joos et al., 2008, JGR; Joos et al., 2014, ACP).

   a. **Response:** This study is based on our previous study (Tully et al., 2022) that showed that using the orographic gravity wave parameterization by Joos et al. (2008, 2014) unrealistically increases ICNC in cirrus when using the P3 ice microphysics scheme (Morrison and Milbrandt, 2015; Dietlicher et al., 2018, 2019). We added a note to this in the text.

   b. **Changes in the text:**

*"Vertical ascent in our model is represented by the updraft, which is calculated as the sum of the grid mean value and a turbulent component represented by the turbulent kinetic energy (Brinkop and Roeckner, 1995). Note that we do not consider orographic effects on the vertical velocity in our model when using the P3 ice microphysics scheme as discussed in Tully et al., (2022a)."*

3. **Comment:** The treatment of pre-existing ice appears to assure the dominance of het which would assure that no cooling from seeding occurs, and that CRE changes must be positive. Therefore, any seeding effect will be a warming effect, as shown in Fig. 3. Nonetheless, this study has value in demonstrating the sensitivity of cirrus properties to seeding, regardless of whether CRE is positive or negative. And it demonstrates the limitations of aircraft seeding. However, in regard to aircraft seeding, it could be mentioned that commercial cloud seeding programs produce AgI seeding aerosol mean diameters on the order of 0.01 μm. Mentioning this would make the r0.01 seeding scenarios appear more realistic.

a. **Response:** Please see our response to your **Comment 1** regarding pre-existing ice. We agree that this issue is still open, so we added an outlook in the discussion section. We are unaware of any commercial applications of AgI seeding with small particles, but we found research on this and included it in the revised text under the Experimental Setup section to justify our small seeding particles.

b. **Changes in the text:**

*"Including such small seeding particles in our model (0.01 $\mu$m) is justified based on previous work on the ice nucleation ability of silve-iodide (AgI). Xue et al. (2013) formulated a parameterization for glaciogenic cloud seeding with AgI in the Weather Research and Forecasting (WRF) model, using a mean particle diameter of 0.04 $\mu$m They reported that the model could reasonably produce the physical processes of cloud seeding. Geresdi et al. (2020) also investigated cloud seeding in the WRF model with slightly larger AgI with a mean diameter of 0.05 $\mu$m and reported that the model also reasonably reproduced the microphysical properties of real clouds. Marcolli et al. (2016) reviewed lab-based experiments of ice nucleation and showed that AgI particles of 20 nm in diameter had an increasing ice nucleation efficiency towards cirrus temperatures (238 K). Finally, Kanji et al. (2017) presented new evidence of the ice nucleation ability of small particles such as pollen and fungal spores, which challenges arguments that only large particles are suitable INPs."*

**Major Comments**

1. **Comment:** Line 275: Please explain the difference between "global mean net top-of-atmosphere (TOA) and net cloud radiative effect (CRE) anomalies". The former accounts for everything, including RH changes, while the latter pertains to clouds only. Many readers may not know this.
   a. **Response:** Good point. We amended the text to make this distinction clear.
   b. **Changes in the text:**

*"Fig. 3 and Tab. 3 present the five-year annual global mean net top-of-atmosphere (TOA) and net cloud radiative effect (CRE) anomalies for each seeding emissions radius and mass scaling factor that we tested. The TOA anomaly refers to the total "all-sky" (Ramanathan, 1987; Wild et al., 2019) radiative effect (i.e., from clouds, aerosols, surface albedo, and changes in atmospheric gases like water vapor), whereas the CRE anomaly refers to the radiative effect of clouds only. The TOA and CRE anomalies scale with the number concentration of seeding particles (Fig 3 and Tab. 3)."*

2. **Comment:** Lines 280-282: The CCT modeling experiment of Gruber et al. (2019, JGR) shows the impact of CCT on lower mixed phase clouds. Do their results support this speculation?
   a. **Response:** Gruber et al. (2019) found seeding led to enhanced riming of cloud droplets, reducing mixed phase cloud cover. We added a brief reference to this in the text.
   b. **Changes in the text:**

*"This latter point is the opposite of what Gruber et al. (2019) found for mixed-phase clouds, which was a reduction in cloud fraction through enhanced riming of cloud droplets onto the ice crystals that formed on injected seeding particles."*

3. **Comment:** Lines 452-3: This appears true for the mid-seeding case but not the low-seeding case.
   a. **Response:** It is true for both cases, but it is insignificant for our low-seeding case, which we allude to further down in the text. However, as that is unclear, we revised the text for greater clarity.
   b. **Changes in the text:**

*"While this signal is somewhat clear for the mid-seeding case, it is unclear for the low-seeding case due to the wide range of the 95% confidence level."*

4. **Comment:** Line 454: Should "Fig. 7d" in this sentence be changed to Fig. 7b?
   a. **Response:** Yes, this should read as 7b. Thank you for pointing this out.

5. **Comment:** Lines 472-474: This explanation makes sense based on other studies, but this study shows ice particle size decreases (and presumably fall speeds as well) with decreasing emission scaling (i.e., decreasing INP concentration). This explanation thus appears to contradict the preceding discussion.
   a. **Response:** We agree. This discussion was reformulated in the revised text.
   b. **Changes in the text:**

*"As shown above, this is the result of new ice crystal formation onto the injected seeding particles, especially for the high-seeding case, which showed ICNC anomalies that exceeded much of the ICNC in the unseeded cirrus. The smaller ice crystals have reduced sedimentation velocities. This is most pronounced in the mid and high-seeding cases, where we find negative IWC anomalies in the lower mixed-phase regime. However, the ice crystal radius anomalies for these two cases are smaller than the anomaly for the low-seeding case due to an increase in IWC because of less efficient sedimentation."*

6. **Comment:** Lines 529-531: Could the use of drones make CCT more viable in this respect, as suggested in Mitchell et al. (2011, Cirrus clouds and climate engineering: New findings on ice nucleation and theoretical basis. In: Planet Earth 2011 - Global Warming Challenges and Opportunities for Policy and Practice, Prof. Elias Carayannis (Ed.), ISBN 978-953-307-733-8, InTech, Available from HYPERLINK "http://www.intechopen.com/artic les/show/title/cirrus-clouds-and-climate-engineering-new-findings-on-ice-nucleation-and- theoretical-basis"). For example, Storelvmo and Herger

(2014) describe a high-latitude seeding approach that would require less flight coverage, and even restricting flights to the Polar Regions would likely result in significant cooling based on their methodology. It seems plausible to increase the density of drone flights in the Polar Regions to address the concerns of this paper. Please comment on this.

    a. **Response:** We had not considered this, but it is a good point to add for future work. We added some discussion on this in the revised text.

    b. **Changes in the text:**

*"Mitchell and Finnegan (2009) proposed that if CCT were implemented in the real-world, a potential delivery mechanism could be to use commercial aircraft, which would have a much less homogeneous spatial extent. Later, Mitchell et al. (2011) also proposed using uncrewed drones for seeding particle delivery, which could significantly enhance public safety but could be much more expensive to operate."*

*"Second, emitting seeding particles from commercial aircraft or from uncrewed drones were proposed as potential delivery mechanisms in the real-world by Mitchell and Finnegan (2009) and Mitchell et al. (2011), respectively. However, aircraft emissions of soot contribute an uncertain effect on cirrus, mostly from uncertainty surrounding the ability of soot to act as an INP (Mahrt et al., 2018, 2020; Lee et al., 2021). In addition, seeding with uncrewed drones could increase the efficiency of potential seeding campaigns by offering dedicated flight paths, but could also be very expensive and associated with legal as well as ethical issues."*

**Technical Comments**

1. **Comment:** Figure 4 caption: There is no mention of the solid and dashed curves shown in these plots; these curves should be defined. They appear to represent the tropopause and the 0°C isotherm.
    a. **Response:** Yes, this was also pointed out by Referee #1. This is fixed in the revised manuscript, including all figures where this is applicable.

2. **Comment:** Figure 7 caption: The y-axis in Fig. 7b appears to indicate microns (change in ice radius) and not temperature as stated in caption.
    a. **Response:** This was also pointed out by Referee #1 and was fixed in the revised manuscript.

3. **Comment:** Line 470: Novemver => November
    a. **Response:** Thank you for pointing out this typo. This was fixed in the revised manuscript.